# Rainbow PO: A Unified Framework for Combining Improvements in Preference Optimization

**Hanyang Zhao**[1*], **Genta Indra Winata**[2*], **Anirban Das**[2*], **Shi-Xiong Zhang**[2],
**David D. Yao**[1], **Wenpin Tang**[1], **Sambit Sahu**[2]
[1]Columbia University     [2]Capital One

## Abstract

Recently, numerous preference optimization algorithms have been introduced as extensions to the Direct Preference Optimization (DPO) family. While these methods have successfully aligned models with human preferences, there is a lack of understanding regarding the contributions of their additional components. Moreover, fair and consistent comparisons are scarce, making it difficult to discern which components genuinely enhance downstream performance. In this work, we propose RAINBOWPO, a unified framework that demystifies the effectiveness of existing DPO methods by categorizing their key components into seven broad directions. We integrate these components into a single cohesive objective, enhancing the performance of each individual element. Through extensive experiments, we demonstrate that RAINBOWPO outperforms existing DPO variants. Additionally, we provide insights to guide researchers in developing new DPO methods and assist practitioners in their implementations. Our code is available at https://github.com/CapitalOne-Research/RainbowPO.

## 1 Introduction

Reinforcement Learning with Human Feedback (RLHF) (Ouyang et al., 2022; Stiennon et al., 2020; Ziegler et al., 2019) has significantly contributed to the success of recently released Large Language Models (LLMs) such as InstructGPT (Ouyang et al., 2022), ChatGPT, and GPT4 (Achiam et al., 2023). However, RLHF is a complex and resource intensive process and requires training a reward model. An alternative to RLHF is Direct Preference Optimization (DPO) (Rafailov et al., 2023) that directly optimizes policies from pairwise preferences by minimizing a supervised learning loss objective, which is viewed as the maximum likelihood estimate for the reward model in RLHF. This approach allows DPO and other DPO variants to bypass the use of RL, resulting in faster speed of end-to-end training and better resource efficiency, while achieving comparable or superior performance to RLHF in downstream tasks such as summarization (Rafailov et al., 2023).

DPO and its success during training foundation models like Llama series (Dubey et al., 2024; Touvron et al., 2023), Mistral (Jiang et al., 2023a), has garnered significant research attention in the LLM alignment space (Winata et al., 2024; Wang et al., 2024b), leading to the development of various extensions. These include variants beyond pairwise ranking, such as Kahneman & Tversky Optimization (KTO, Ethayarajh et al. (2023)) and MallowsPO (Chen et al., 2024a), unified perspectives on loss parameterization, such as Identity Preference Optimization (IPO, Azar et al. (2024)) and Generalized Preference Optimization (GPO, Tang et al. (2024)), distribution correction methods like Rejection Sampling Optimization (RSO, Liu et al. (2023)), and reference model-free alternatives, such as Contrastive Preference Optimization (CPO, Xu et al. (2024)), Odds Ratio Preference Optimization (ORPO, Hong et al. (2024)), and Simple Preference Optimization (SimPO, Meng et al. (2024)). Each of these DPO variants claims to outperform the original DPO in downstream task evaluations by introducing specific additional components, or mathematically modifying the loss objective. In the rest of the paper, we will refer DPO variants collectively as xPOs for simplicity.

Comparing these xPOs proposed in the literature is not always straightforward due to differences in the base model size and architecture, the alignment datasets, the experimental setup as well as the

---

*Equal Contribution. Contacts: hz2684@columbia.edu, genta.winata@capitalone.com, anirban.das3@capitalone.com.

evaluation metrics. Subsequently, it becomes difficult to assess the effectiveness and choose among different xPO methods given a problem. A brute force comparison across all existing methods is prohibitively expensive and inefficient. Therefore, it is crucial that we study the performance characteristics of each proposed method in the literature by evaluating the xPOs' performances under at least one convincing and representative setup. Further, despite the success of the xPO family, a fundamental question remains unexplored:

*What are the components proposed in* xPOs *that actually improve the performance over* DPO*?*

Surprisingly, there is still a lack of comprehensive work studying the progress in the literature and summarizing the core practical components of these methods that lead to improvement over DPO. To demystify the reasons for their effectiveness, we hypothesize that the main benefits of these methods stem from the combination of several mathematically orthogonal effective components. In this paper, we validate our hypothesis by decomposing the xPOs and identifying these orthogonal components upon DPO. We further assess their effectiveness through downstream task evaluations, ruling out the components that do not contribute to performance improvements. Given these orthogonal identified beneficial components for preference optimization, a natural question arises:

*Can these individual components complement each other and be effectively combined?*

Our question is largely motivated by the previous study RAINBOW (Hessel et al., 2018) that explored improvements over Deep Q-Networks algorithm (DQN) (Mnih et al., 2015) in traditional Reinforcement Learning (RL). The summarization and comparsion in Hessel et al. (2018) greatly enhances the understanding for improving DQN, and the resulting algorithm Rainbow, still serves as a benchmark (Raffin et al., 2021). However, such a study for RLHF is still underexplored. This shows a gap in the literature, that elicits an answer to the question of combining different xPO extensions evaluated in a comprehensive setting. To bridge this gap, we propose RAINBOWPO, a unified framework that integrates existing xPOs' components, and deploys useful and essential components in a principled manner to achieve better performance. To conclude, our contributions in this paper are as follows:

(1) We conduct a comprehensive study on more than 10 offline representative variants of DPOs (xPOs) from a *practical* aspect by analyzing their loss functions for optimization. We conclude several mathematically orthogonal directions along which these methods propose to optimize over the original DPO loss, analyze the usefulness of each method theoretically and empirically, and provide comparisons under the same representative setup.

(2) We identify and summarize 7 broad components across all DPO extensions: length normalization, link function, margin / home advantage, reference policy, contextual scaling, rejection sampling optimization (RSO), and supervised fine-tuning (SFT) loss, and justify that *four* of them are effective through extensive hyper-parameters search, model training and evaluations. Additionally, we also propose a better way of formulating the reference policy by mixing the SFT policy with a margin (see details in the reference policy of Section 3.1), and demonstrate the advantage of this approach over using just SFT policy (in DPO) or just margin (in SimPO).

(3) Finally, we propose RAINBOWPO, a DPO variant that combines three essential and orthogonal components from existing xPOs. Combining other adjustment on optimization hyper-parameters, we show that our algorithms perform the best among all open-sourced algorithms when tuning Llama3-8B-Instruct. In the widely adopted LLM benchmark Alpaca-Eval2 [1], RAINBOWPO improves Llama3-8B-Instruct from 22.92% to 51.66% for Length Controlled Win Rate (LC WR), with just access to a reward model to form the offline preference dataset and no further online sampling. We also perform an ablation study and show that all adopted elements in RAINBOWPO are indeed necessary to achieve the best result.

**Related Work**. Below we provide a (non-exhaustive) list of other relevant references to this work.

Compared to human feedback in original RLHF, existing works have improved the scalability by utilizing AI feedback (Bai et al., 2022; Lee et al., 2023). For such need of constructing better AI feedback, recent works also proposed various reward models for formulating better preference datasets, like PairRM (Jiang et al., 2023b), ArmoRM (Wang et al., 2024a), RRM (Liu et al., 2024a), and RM benchmarks like Reward Bench (Lambert et al., 2024).

---

[1] https://github.com/tatsu-lab/alpaca_eval.

We also find works that target at understanding DPO methods related to our work. Liu et al. (2024b) studies the effect of reference policy in the preference optimization; Saeidi et al. (2024) compare the performance of DPO, IPO, CPO, KTO for tuning Mistral 7B (Jiang et al., 2023a) based models, and mainly studied the roles of SFT stage for alignment methods.

The rest of the paper is organized as follows. We provide backgrounds on RLHF and DPO in Section 2. In Section 3, we summarize the current directions in existing xPOs and the development of RAINBOWPO, followed by detailed experimental results in Section 4. Finally, we present our conclusion in Section 5.

## 2 PRELIMINARIES AND MOTIVATION

In this section, we first briefly introduce RLHF and DPO as the foundation method, and then discuss on extensions of DPO (xPOs) to understand what are the components proposed in the literature.

RLHF starts with fine-tuning a pre-trained large language model by supervised learning on high-quality data for some downstream tasks of interest (e.g., dialogue, summarization, etc.), to acquire a model $\pi^{\text{SFT}}$. This step is referred to as the SFT phase. For instance, for training Instruct-GPT (Ouyang et al., 2022), GPT-3 (Brown et al., 2020) is first fine-tuned on the given input prompt distribution. The second stage of RLHF is known as reward modeling, i.e., researchers collect preferences $\mathcal{D} = (x, y_w, y_l)$ on the generations of fine-tuned model $\pi^{\text{SFT}}$, and learns a reward model $r^*(x, y)$ that could represent the quality or the rating of generation $y$ with respect to prompt $x$. The final step is policy optimization on $\pi_{\text{SFT}} = \pi_{\text{ref}}$, by maximizing a regularized reward to obtain the optimal policy model $\pi^*$ through reinforcement learning:

$$\max_{\theta} \mathbb{E}_{x \sim \mathcal{D}} \left[ \mathbb{E}_{y \sim \pi_{\theta}(y|x)} \left[ r^*(x, y) \right] - \beta \mathrm{KL} \left( \pi_{\theta}(\cdot \mid x) \| \pi_{\text{ref}}(\cdot \mid x) \right) \right], \quad (1)$$

in which $\beta > 0$ denotes the regularization constant. For ease of reference, we prvide more detailed description of RLHF in Appendix A.1, and we also add a table of notations in Table 7 in Appendix B.

### 2.1 DIRECT PREFERENCE OPTIMIZATION (DPO)

One disadvantage of RLHF is that the RL step often requires substantial computational effort (e.g., to carry out PPO). The idea of DPO is to combine the reward model and RL in RLHF into a single objective, bypassing the computation in the RL step. Given the same preference pairs $\mathcal{D} = (x, y_w, y_l)$ utilized for reward modeling in RLHF, the DPO objective yields:

$$\min_{\theta} \mathcal{L}_{\text{DPO}} \left( \pi_{\theta}; \pi_{\text{ref}} \right) := -\mathbb{E}_{(x, y_w, y_l) \sim \mathcal{D}} \left[ \log \sigma \left( \beta \log \frac{\pi_{\theta} \left( y_w \mid x \right)}{\pi_{\text{ref}} \left( y_w \mid x \right)} - \beta \log \frac{\pi_{\theta} \left( y_l \mid x \right)}{\pi_{\text{ref}} \left( y_l \mid x \right)} \right) \right], \quad (2)$$

where $\sigma(\cdot)$ is the sigmoid function and $\beta$ is a regularization parameter similar to the one in RLHF. DPO thus yields a supervised learning problem, and requires much less computation than the RL based RLHF. The objective in Equation 2 can be understood as maximizing the likelihood difference between the preference pairs under the policy, encouraging the model to more likely generate the preferred answers than non-preferred. We refer more details of DPO derivation in Appendix A.2.

### 2.2 MOTIVATION: REVISITING XPOS

Since DPO is proposed, there is huge interest in developing and improving DPO, leading to numerous xPOs. Different xPOs can be motivated by theoretical concerns like relaxing or extending preference distribution assumptions in IPO and MallowsPO, human aware loss function in KTO, or from practical aspects like reference model-free alternatives, like CPO, ORPO and SimPO. We provide an non-exhaustive list in Table 8 in Appendix B for the ease of revisiting and comparison.

Despite their differing motivations, xPOs share a primary objective to optimize. We thus take the loss objectives as the first class citizen, and mathematically understand the parts that are commonly adopted or differ in xPOs. Before going into detailed categorization, we want to first argue that, in existing preference optimization literature, there is a lack of comprehensive studies on revisiting and examining DPO variants in their mathmatical objectives. As a consequence, some papers may have implicitly proposed some designs for improvement and even didn't highlight it. As a motivating example, we revisit ORPO objective, which proposes to maximize an odd ratio difference (for an

event A with probability $p$, the odds ratio is defined as $p/(1-p))$ between the winning and losing answers:

$$\mathcal{L}_{\text{ORPO}}(\pi_\theta) = -\mathbb{E}[\underbrace{\log p_\theta(y_w|x)}_{\mathcal{L}_{\text{SFT}}(\pi_\theta)} + \underbrace{\lambda \log \sigma \left( \log \frac{p_\theta(y_w|x)}{1-p_\theta(y_w|x)} - \log \frac{p_\theta(y_l|x)}{1-p_\theta(y_l|x)} \right)}_{\lambda \cdot \mathcal{L}_{\text{PO}}(\pi_\theta)}], \quad (3)$$

in which the expectation is for $(x, y_w, y_l) \sim \mathcal{D}$, and $p_\theta(y|x) = \exp\left(\frac{1}{|y|} \log \pi_\theta(y|x)\right)$. Rewriting terms in $\mathcal{L}_{\text{PO}}(\pi_\theta)$, we could derive a upper bound as (see proof in Appendix D.1):

$$\mathcal{L}_{\text{PO}}(\pi_\theta) \leq -\log \sigma(\frac{1}{1-p_\theta(y_l|x)} \underbrace{\left( \frac{1}{|y_w|} \log \pi_\theta(y_w|x) - \frac{1}{|y_l|} \log \pi_\theta(y_l|x) \right)}_{\Delta_\theta}) := \bar{\mathcal{L}}_{\text{PO}}(\pi_\theta), \quad (4)$$

if assuming $\Delta_\theta > 0$ for all $x$. The upper bound $\bar{\mathcal{L}}_{\text{PO}}$ is sharp, as $\bar{\mathcal{L}}_{\text{PO}} - \mathcal{L}_{\text{PO}} = \mathcal{O}(\Delta_\theta)^2$; thus minimizing ORPO loss could be understood as *reference-model free* DPO with *length normalization* (namely $1/|y_w|$ and $1/|y_l|$, see more detailed explanation in Section 3) and a *contextual dependent* $\beta(x) = 1/(1-p_\theta(y_l|x))$. Length normalization is one of the key ideas adopted in SimPO:

$$\mathcal{L}_{\text{SimPO}}(\pi_\theta; \gamma) := -\mathbb{E}_{(x, y_w, y_l) \sim \mathcal{D}} \left[ \log \sigma \left( \frac{\beta}{|y_w|} \log \pi_\theta(y_w|x) - \frac{\beta}{|y_l|} \log \pi_\theta(y_l|x) - \gamma \right) \right], \quad (5)$$

which is evident in SimPO objective, though proposed after ORPO. This connection is however unaware by the literature. This thus calls for a comprehensive analysis of the contributed elements in different xPOs so far, as many methods may overlap in contributed directions without awareness, and bringing this out right away could possibly prevent repetitive work or efforts in the future.

Following similar analysis of different representative XPO methods for pairwise preferences, including DPO (Rafailov et al., 2023), IPO (Azar et al., 2024), CPO (Xu et al., 2024), GPO (Tang et al., 2024), RSO (Liu et al., 2023), ODPO (Amini et al., 2024), ORPO (Hong et al., 2024), MallowsPO (Chen et al., 2024a), SimPO (Meng et al., 2024), we come up with seven broad categories, which is able to explain most popular DPO variants in the literature, as in Table 1, This provides a straightforward illustration of the main ideas and connections of existing methods. The meanings and details of the categories are elaborated in Section 3.

| Method | Length Norm. | Link Func. | Home Adv. | Ref. Policy | Contextual Scaling | RS | SFT Loss |
|---|---|---|---|---|---|---|---|
| DPO | × | logistic | × | SFT | × | × | × |
| SLiC-HF | × | hinge | × | SFT | × | × | ✓ |
| IPO | × | square | × | SFT | × | × | × |
| CPO | × | logistic | × | Free | × | × | ✓ |
| RSO | × | logistic / hinge | × | SFT | × | ✓ | × |
| ODPO | × | logistic | ✓ | SFT | × | × | × |
| ORPO | ✓ | logistic | × | Free | implicitly | × | ✓ |
| WPO | × | logistic | × | SFT | ✓ | × | × |
| MallowsPO | × | logistic | × | SFT | ✓ | × | × |
| SimPO | ✓ | logistic | ✓ | Free | × | × | × |
| RainbowPO | ✓ | logistic | × | mixing | ✓ | × | × |

Table 1: Mapping of xPOs with mathematically orthogonal components (see more details in Appendix C) and validation results of their effectiveness by the downstream task evaluations.

# 3 RAINBOWPO: A UNIFIED FRAMEWORK

## 3.1 COMPONENT DESCRIPTIONS

We first explain in detail about the components we categorized, after which we propose a generic framework, which we name as RainbowPO, to combine these components.

**Length Normalization.** The literature has noticed a verbosity issue of DPO aligned models, as the aligned model may generate answers significantly longer than both preferred and rejected answers (Park et al., 2024). This also could lead to an inflated win wate when evaluating model performance, as LLM-as-a-judge can be susceptible to length bias: Wang et al. (2023) has noticed that when

evaluating 13B parameter models in head-to-head comparisons with the Davinci-003 model, win rates have a strong correlation (0.96) with the average number of unique tokens in the model's response. To address this issue, one promising direction noticed in the literature is to incorporate explicit length penalties, like in R-DPO (Park et al., 2024) and SimPO (Meng et al., 2024):

$$r_\theta^{\text{LR}}(x,y) = r_\theta(x,y) - \alpha|y|, \text{ and } r_\theta^{\text{LN}}(x,y) = \frac{1}{|y|} r_\theta(x,y), \quad (6)$$

in which $r_\theta(x,y) = \log \frac{\pi_\theta(y|x)}{\pi_{\text{ref}}(y|x)}$ is the implicit reward model (Rafailov et al., 2023). From an optimization perspective, maximization with respect to $r_\theta^{\text{LN}}(x,y)$ is equivalent to $r_\theta^{\text{LR}}(x,y)$ with a specific $\alpha$ (might be prompt $x$ dependent). Directly applied to DPO, the resulting objective yields:

$$\mathcal{L}_{\text{LN-DPO}}(\pi_\theta; \pi_{\text{ref}}) := -\mathop{\mathbb{E}}_{(x,y_w,y_l)\sim\mathcal{D}} \log \sigma \left( \frac{\beta}{|y_w|} \log \frac{\pi_\theta(y_w \mid x)}{\pi_{\text{ref}}(y_w \mid x)} - \frac{\beta}{|y_l|} \log \frac{\pi_\theta(y_l \mid x)}{\pi_{\text{ref}}(y_l \mid x)} \right). \quad (7)$$

Why length normalization could help prevent the verbosity issues can be explained through examining the gradient of the loss respectively: $\nabla_\theta \mathcal{L}_{\text{LN-DPO}}(\pi_\theta; \pi_{\text{ref}}) =$

$$-\beta \mathbb{E}\left[ \sigma\left( r_\theta^{\text{LN}}(x,y_l) - r_\theta^{\text{LN}}(x,y_w) \right) \left( \frac{1}{|y_w|} \nabla_\theta \log \pi_\theta(y_w \mid x) - \frac{1}{|y_l|} \nabla_\theta \log \pi_\theta(y_l \mid x) \right) \right], \quad (8)$$

thus the gradient of length normalized DPO can be understood as taking a discount factor $\frac{1}{|y_w|}$ of the length for longer sequence. We also empirically justify the effectiveness of length normalization by comparing to the vanilla DPO trained models, and witness the consistent smaller average length, independent of the regularization constant $\beta$. See results of average length in Section 4.

**Link Function.** SLiC-HF (Zhao et al., 2023) and GPO (Tang et al., 2024) both realized that the DPO objective could be understood as taking $f$ (we refer this as *link function*) as $-\log \sigma(\cdot)$ in:

$$\mathcal{L}_{\text{GPO}} = \mathop{\mathbb{E}}_{(x,y_w,y_l)\sim\mathcal{D}} \left[ f\left( \beta \log \frac{\pi_\theta(y_w|x)}{\pi_{\text{ref}}(y_w|x)} - \beta \log \frac{\pi_\theta(y_l|x)}{\pi_{\text{ref}}(y_l|x)} \right) \right], \quad (9)$$

thus unifying DPO, IPO, SLiC (without SFT loss) as instances of taking $f$ as logistic $-\log \sigma(\cdot)$, square $(x - 1/2)^2$ or hinge function $\max(0, \delta - x)$ for some margin $\delta > 0$ (see more details in Appendix C), separately. For identifying the best link function, we did an exclusive parameter search for DPO and IPO separately, however we found DPO, i.e. adopting $-\log \sigma(\cdot)$ as the link function, empirically performs better than IPO evaluated by AlpacaEval2, despite the weaker theoretical assumption of preferences in IPO. Thus we stick to the link function to be $-\log \sigma(\cdot)$ in this paper.

**Home Advantage / Margin.** In SLiC, IPO, ORPO, SimPO, there exists a term which also targets at encouraging the difference between the reward model difference. It is also referred in SimPO as the term of home advantage $\gamma$ (the terminology comes from an extension of the vanilla Bradley-Terry Model): $\text{logit}(\text{Prob}(i \text{ beats } j)) = r_i - r_j - \gamma$. Thus the likelihood could be written as:

$$p^*(y_1 \succ y_2 \mid x) = \sigma(r^*(x,y_1) - r^*(x,y_2) - \gamma), \quad (10)$$

which takes the losing prompt in a home advantage when $\gamma > 0$. SimPO shows the effectiveness of this margin under the reference-free setup; however, when we adopt the margin for vanilla DPO (i.e. with the reference policy) with the optimal $\beta$, we do not witness an increase of the performance when adjusting the margin, either further adopting DPO with length normalization or not. In Figure 1a, the performance steadily decreases when increasing the margin $\gamma$ in DPO, i.e. we optimize:

$$\mathcal{L}_{\text{DPO+}}(\pi_\theta; \pi_{\text{ref}}, \gamma) := -\mathbb{E}_{(x,y_w,y_l)\sim\mathcal{D}} \left[ \log \sigma\left( \beta \log \frac{\pi_\theta(y_w \mid x)}{\pi_{\text{ref}}(y_w \mid x)} - \beta \log \frac{\pi_\theta(y_l \mid x)}{\pi_{\text{ref}}(y_l \mid x)} - \gamma \right) \right], \quad (11)$$

This questions the true explanation about the effectiveness of the margin term in SimPO. We provide the answer as understanding margin as a reference policy right in the next paragraph.

**Reference Policy.** DPO takes the SFT policy as the reference policy motivated by the standard RLHF pipeline. However, recently proposed methods like CPO, ORPO, and SimPO (Xu et al., 2024; Hong et al., 2024; Meng et al., 2024) all suggested a reference-free objective could yield

the same or even better performance. CPO and ORPO further utilized an extra SFT loss to force regularization, while for SimPO, such regularization is not enforced. Given our prior examination that home advantage can hardly improve over DPO, we argue that *the margin term in* SimPO loss Eq. 5 *should be understood as a term for "reference policy" instead of the "home advantage".*

Concretely, we could hypothesize that there exists a "good policy" $\pi_\gamma$ such that, for each prompt and preference pairs in the dataset, the normalized log likelihood ratio of preferred response to non-preferred response is a positive constant, which we denote as $\pi_\gamma$. We assume that $\pi_\gamma$'s normalized implicit reward model is perfect at in-distribution pairwise classification and yields $\frac{\pi_\gamma(y_w|x)^{1/|y_w|}}{\pi_\gamma(y_l|x)^{1/|y_l|}} = \exp(\gamma)$ for any $x$. If so, the loss of SimPO (defined as in Equation 5) could be rewritten as:

$$\mathcal{L}_{\text{SimPO}}\left(\pi_\theta; \gamma\right) \equiv - \mathop{\mathbb{E}}_{(x,y_w,y_l)\sim\mathcal{D}} \log \sigma \left( \frac{\beta}{|y_w|} \log \frac{\pi_\theta\left(y_w \mid x\right)}{\pi_\gamma\left(y_w \mid x\right)} - \frac{\beta}{|y_l|} \log \frac{\pi_\theta\left(y_l \mid x\right)}{\pi_\gamma\left(y_l \mid x\right)} \right). \quad (12)$$

This transformation motivates us to further propose a new mechanism which we call as *mixing (reference) policy*. If taking $\pi_{\text{sft}}$ as the reference policy is too conservative (not strong enough), and taking $\pi_\gamma$ policy can help improve the performance but totally neglects the original SFT model implicit preference information, can we benefit from a mixing of these two policies? The answer is YES. Consider a exponential mixing of likelihood of reference policy $\pi_\gamma$ and $\pi_{\text{ref}}$ (we use $\pi_{\alpha,\gamma}$ to denote the relevance of the resulting reference policy to both $\alpha$ and $\gamma$) with $\alpha \in [0,1]$, defined as:

$$\pi_{\alpha,\gamma}(y \mid x) \propto \pi_{\text{ref}}^\alpha(y \mid x) \cdot \pi_\gamma^{1-\alpha}(y \mid x). \quad (13)$$

Then if we use $\pi_{\alpha,\gamma}$ as the reference policy in (7), we yield $\mathcal{L}_{\text{LN-DPO}}(\pi_\theta; \pi_{\alpha,\gamma})$ with a practical form:

$$- \mathop{\mathbb{E}}_{(x,y_w,y_l)\sim\mathcal{D}} \log \sigma \left( \beta \log \frac{\pi_\theta\left(y_w \mid x\right)^{1/|y_w|}}{\pi_\theta\left(y_l \mid x\right)^{1/|y_l|}} - \alpha\beta \log \frac{\pi_{\text{ref}}\left(y_w \mid x\right)^{1/|y_w|}}{\pi_{\text{ref}}\left(y_l \mid x\right)^{1/|y_l|}} - (1-\alpha)\gamma \right). \quad (14)$$

Notice that $\mathcal{L}_{\text{LN-DPO}}(\pi_\theta; \pi_{\alpha,\gamma}) = \mathcal{L}_{\text{LN-DPO}}(\pi_\theta; \pi_{0,\gamma}) = \mathcal{L}_{\text{SimPO}}(\pi_\theta; \gamma)$, thus SimPO is an instance of mixing policy by taking $\alpha = 0$; $\mathcal{L}_{\text{LN-DPO}}(\pi_\theta; \pi_{1,\gamma}) = \mathcal{L}_{\text{LN-DPO}}(\pi_\theta)$, thus $\alpha = 1$ corresponds to DPO applied with length normalization as in Equation 7. For finding a good $\pi_{\alpha,\gamma}$, we first take $\alpha = 0$ and tune the best $\gamma$, which is similar to SimPO; we then afterwards tune $\alpha$ using obtained $\gamma$.

According to our experiment results, we indeed find that there exists $\alpha \in (0,1)$ that performs better than both sides (i.e. $\alpha = 0$ or $\alpha = 1$), see Figure 1b. Recent work (Liu et al., 2024b) analyze the role of reference model, and argue that stronger reference model could benefit DPO; our finding is consistent, as we further explicitly design a choice of better reference model for better performance.

**Rejection Sampling.** Since the proposal of DPO, there is controversy on the exact equivalence of DPO and RLHF. RSO (Liu et al., 2023), further pointed out that the data should be generated from the optimal policy if treating DPO objective as maximum likelihood estimation. Thus RSO adopts a statistical rejection sampling for sampling preference dataset $\mathcal{D}$ generated by the optimal policy to mitigate this distribution difference in DPO.

To address the intrinsic different variance schedules of reward model for different prompts, and stablize the process for formulating the preference dataset, we also adopt a modified version of RSO by computing the percentile reward (or the ranking reward) in the whole generation set instead of utilizing the true reward, which we found that can stabilize the generation and yield better results when further applied with DPO, as in Algorithm 1.

Similar to in RSO (Liu et al., 2023), we search the best temperature hyper-parameter for RSO through the downstream task performance as the validation metric, which we detail in Figure 1c. We then use the empirically best performed temperature constant $\tau$ to formulate the preference dataset as $\mathcal{D}_{\text{RS}}$.

---

**Algorithm 1** RS$^+$ for preferences formulation.

---

For each prompt $x$, start with an empty set $\mathcal{Y} \leftarrow \{\}$.
Generate $N \gg M$ answers $y_i \sim \pi_{\text{sft}}(y \mid x)$, for $i \leq N$ as candidates.
Compute each $y_i$'s percentile $\mathcal{P}_i(x)$ based on $r(x, y_i)$ over the whole $N$ answers for prompt $x$.
Initialize counting number $j = 0$.
**while** $|\mathcal{Y}| < M$ **do**
    $j = j + 1$ and generate $u \sim U[0,1]$
    **if** $u \leq \exp((\mathcal{P}_i - 1)/\tau)$ **then**
        Accept $y_i$ and add it to $\mathcal{Y}$.
    **else**
        Reject $y_i$.
    **end if**
**end while**
Let $y_w = \arg\max_{y \in \mathcal{Y}} r(x, y)$
Let $y_l = \arg\min_{y \in \mathcal{Y}} r(x, y)$

---

**Contextual Scaling.** Existing work also considered the contextual difference: some preference pairs might be of higher uncertainty or have more *dispersion*. In this paper, we adopt the idea of MallowsPO in Chen et al. (2024a) by introducing a contextual scaling factor $\phi(x)$ on the likelihood difference, which yields:

$$\mathcal{L}_{\text{MallowsPO}}(\pi_\theta; \pi_{\text{ref}}) = - \underset{(x,y_w,y_l)\sim\mathcal{D}}{\mathbb{E}} \log \sigma \left( \phi(x) \left[ \beta \log \frac{\pi_\theta(y_w|x)}{\pi_{\text{ref}}(y_w|x)} - \beta \log \frac{\pi_\theta(y_l|x)}{\pi_{\text{ref}}(y_l|x)} \right] \right), \quad (15)$$

if adding this factor to DPO objective. The scaling factor is motivated by the Mallows ranking model which has a natural carrier of dispersion index. In MallowsPO, $\phi(x)$ corresponds to a normalized predictive entropy of the preference pair $(x, y_w, y_l)$ ($y_{w,i}$ and $y_{l,i}$ denote the $i^{\text{th}}$ component of $y_w$ and $y_l$):

$$\phi(x) = - \log \left( \frac{\sum_{i=1}^{N-1} \left[ H_{\pi_{\text{ref}}} \left( Y_{i+1} \mid Y_i = y_{w,i} \right) + H_{\pi_{\text{ref}}} \left( Y_{i+1} \mid Y_i = y_{l,i} \right) \right]}{2 \log n} \right). \quad (16)$$

**SFT Loss.** SFT loss is straightforward by adding extra SFT loss term on the winning answer, or a reference answer (for regularization, which appears for reference-free methods like CPO and ORPO). For example, if adding SFT for DPO with preferred/winning answers, we yield:

$$\mathcal{L}_{\text{SFT}} \left( \pi_\theta; \lambda \right) := -\mathbb{E} \left[ \log \sigma \left( \beta \log \frac{\pi_\theta \left( y_w \mid x \right)}{\pi_{\text{ref}} \left( y_w \mid x \right)} - \beta \log \frac{\pi_\theta \left( y_l \mid x \right)}{\pi_{\text{ref}} \left( y_l \mid x \right)} \right) + \lambda \log(\pi_\theta \left( y_w \mid x \right)) \right], \tag{17}$$

However, we find that adding SFT loss could largely degrade the performance, as in Section 4.

### 3.2 UNIFIED FORMULATION: RAINBOW

Combining the advances proposed above, we propose a following preference optimization objective, for which we refer our method as RAINBOWPO:

$$\mathcal{L}_{\text{RAINBOWPO}} \left( \pi_\theta; \pi_{\text{ref}} \right) := - \underset{(x,y_w,y_l)\sim\mathcal{D}^*}{\mathbb{E}} f \left[ \phi(x) \left( \frac{\beta}{|y_w|^\eta} \log \frac{\pi_\theta \left( y_w \mid x \right)}{\pi_{\alpha,\gamma} \left( y_w \mid x \right)} - \frac{\beta}{|y_l|^\eta} \log \frac{\pi_\theta \left( y_l \mid x \right)}{\pi_{\alpha,\gamma} \left( y_l \mid x \right)} \right) \right], \tag{18}$$

in which $\eta \in \{0, 1\}$, and $\pi_{\alpha,\gamma}$ is as defined in Equation (13). The preference dataset $\mathcal{D}^*$ can be $\mathcal{D}_{\text{RS}}$, which means that the preference dataset is formulated by by rejection sampling from the original dataset's prompts as mentioned in Algorithm 1, if we have access to a reward model's true value. If the reward model is black-box oracle, namely we cannot access the true reward value, we will always utilize the usual formulation way of preference dataset $\mathcal{D}$, detailed in the experiment section.

**Hyper-parameters**. Like all other xPOs, to achieve the best performance, RainbowPO can introduce an extensive amount of hyper-parameter search for the best performing $f$, $\alpha$, $\beta$, $\gamma$ and whether $\eta = 1$. For efficient hyper-parameter search, we conducted a greedy search method with the help of our framework and decomposition of effective elements: we search for the best hyper-parameters for those that affects the performance in the most when we gradually add designs to the preference optimization methods. For example, when adding length normalization to the methods, we only search for the best hyper-parameter for the regularization parameter $\beta$, and will fix the learning rate and all the training args, which prevents the parameter searching space from exploding.

## 4 EXPERIMENTS

To evaluate the performance of the xPOs algorithms, we conducted extensive experiments on training models with various xPOs configurations and compared their instruction-following capabilities.

**Experimental Setup.** We choose Llama3-8B-Instruct[2] as our model base to fine tune, mainly because that aligning this widely adopted and flagship instruct model is of great interest to the whole community and meets the standard as a representative setup for alignment. It can also help mitigate the uncertainty from probably not perfectly supervised fine-tuned models.

For evaluation metric, we use widely adopted benchmark AlpacaEval2, which is composed of 805 questions and evaluate the instruction following capability of the model. AlpacaEval2 evaluates

---

[2]https://huggingface.co/meta-llama/Meta-Llama-3-8B-Instruct.

models with the win rates (WR) of model generations against the reference/base answers generated by `GPT4-turbo`. The comparisons are by default annotated by a `GPT4-turbo` and the resulting WR has a 68.5% consistency with human evaluation, according to official AlpacaEval2 website. Length Controlled (LC) Win Rate (WR) is a debiased version of the WR that control for the length of the outputs and increase the WR's correlation to Chat Arena,[3] while significantly decreasing the length gameability of the annotator. To cross validate the effectiveness of the model and mitigate possible bias of `GPT4`, we also adopt Llama3-70B instruct as the judge, which is reported to have a 67.5% win rate consistency to humans (close to the performance of `GPT4`). We include more detailed background information of AlpacaEval2 in Appendix E.1.

For formulating the preference dataset $\mathcal{D}$, we follow the standard RLHF pipeline by directly generating answers from the model (which is thus an on-policy dataset, but the algorithm is still offline) and get AI feedbacks as in SimPO (Meng et al., 2024): we generate 5 answers from Llama3-8B-Instruct for each prompt in UltraFeedback (Cui et al., 2023), rank them with scores evaluated by ArmoRM (Wang et al., 2024a), and choose the best/worst one as winning/losing answer to form the preference pairs. For training, we adopted the popular library Transformer Reinforcement Learning (TRL),[4] which already implemented most aforementioned xPOs algorithms and make everything under the same backend and easy to reproduce. If not specified, we train the model with 3 training epochs, which typically yields better performance for each xPOS according to our replication.

## 4.1 Effectiveness of Different Components

**Individual Components Results.** We first study the effectiveness of adding individual components. We use + to denote that only the component(s) after is added to DPO baseline as in Table 2. From the results, we could notice that some components may not provide firm improvement over the baseline, no matter being added individually or combined. For example, for home advantage, we tune different values under the best performed $\beta$ for DPO, and also always witness a degradation in performances, see Figure 1a. For link function, we examine the square loss in IPO and did not see performance gain over the DPO baseline. Other components (LN, Mixing reference policy, CS) indeed help improve the metric even added individually. Compared to SimPO, using mixing reference policy yields also better results as in Figure 1b. The average WR gain is reported in the last column.

| Models | AlpacaEval2 (GPT4) | | | | AlpacaEval2 (Llama3-70B) | | | | |
|---|---|---|---|---|---|---|---|---|---|
| | LC WR (%) | Δ (%) | WR (%) | Δ (%) | LC WR (%) | Δ (%) | WR (%) | Δ (%) | Avg. Δ (%) |
| Base model | 41.88 | - | 42.29 | - | 57.78 | - | 57.96 | - | - |
| + Length Norm. (LN) | 44.27 | 2.39 | 42.37 | 0.08 | 61.37 | 3.59 | 58.94 | 0.98 | +1.76 |
| + Ref. Policy Mixing (Mix) | 40.18 | -1.7 | 41.25 | -1.04 | 60.67 | 2.89 | 57.95 | -0.01 | +0.04 |
| + Contextual Scaling (CS) | 41.14 | -0.74 | 41.44 | -0.85 | 60.06 | 2.28 | 57.90 | -0.06 | +0.16 |
| + Link Function (LF) | 39.53 | -2.35 | 39.07 | -3.22 | 58.13 | 0.35 | 56.34 | -1.62 | -1.21 |
| + Home Advantage (HA) | 41.70 | -0.18 | 39.85 | -2.44 | 59.01 | 1.23 | 56.41 | -1.55 | -0.74 |
| + Rejection Sampling (RSO) | 42.87 | 0.99 | 42.50 | 0.21 | 58.86 | 1.08 | 56.02 | -1.94 | +0.09 |
| Base model + LN | 44.27 | - | 42.37 | - | 61.37 | - | 58.94 | - | - |
| + LN + Mix | **47.45** | 3.18 | **45.89** | 3.52 | 61.91 | 0.54 | 58.07 | -0.87 | +1.59 |
| + LN + CS | 45.92 | 1.65 | 42.36 | -0.01 | 61.88 | 0.51 | 58.20 | -0.74 | +0.35 |
| + LN + HA | 42.77 | -1.50 | 41.38 | -0.99 | 60.99 | -0.38 | 59.78 | 0.84 | -0.51 |
| + LN + RS | 43.22 | -1.05 | 41.96 | -0.41 | 61.03 | -0.34 | 57.02 | -1.92 | -0.93 |
| + LN + SFT Loss | 39.90 | -4.37 | 38.66 | -3.71 | 60.42 | -0.95 | 58.94 | 0.00 | -2.26 |

Table 2: Model performance results on each component after model training for 3 epochs.

**Components Combination Results.** Given the effectiveness of length normalization, we further test the combination of LN and other components. We do find that mixing policy could help improve the performance much more remarkablly when combined with LN-DPO than just DPO: it provides a 1.6% win rate extra gain compared to only 0.04% on DPO. However, we found that the RSO can improve DPO, but will yield worse performance when applied with length normalization. Thus, we do find that despite that these elements are apparently mathematically orthogonal, they are *not empirically independent*. Given the positive results and effectiveness of length normalization, mixing reference policy and contextual scaling, we propose RainbowPO, as the combination of these three elements. We then examine the effectiveness of our method and each elements by gradually com-

---

[3] `https://lmarena.ai/`.
[4] `https://huggingface.co/docs/trl/index`.

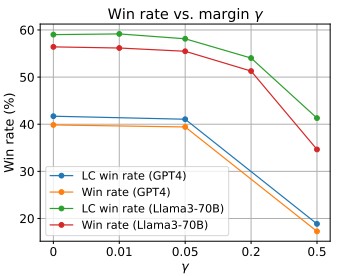 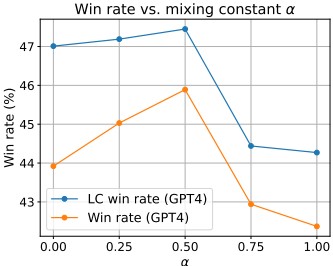 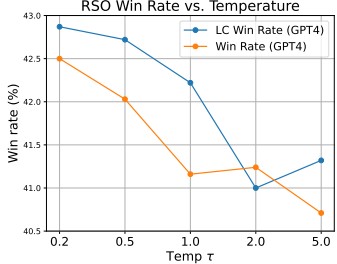

(a) Effects of Home Advantage / Margin.  (b) Effects of Reference Policy Mixing.  (c) Performance Difference for different temperature $\tau$ in RSO.

Figure 1: Dynamics of changing home adv., reference policy mixing and different temp. in RSO.

bining the elements one by one and greedy search of the best hyper-parameters. We finally achieve a 51.66% win rate for AlpacaEval2, surpassing the GPT4-1106 preview (see details in Appendix E.2).

| Models RainbowPO | AlpacaEval2 (GPT4) | | | | AlpacaEval2 (Llama3-70B) | | |
|---|---|---|---|---|---|---|---|
| | LC WR (%) | $\sigma$ | WR (%) | $\sigma$ | LC WR (%) | WR (%) | avg length ($\downarrow$) |
| Base model | 41.88 | 0.77 | 42.29 | 1.46 | 57.78 | 57.96 | 2,169 |
| $\oplus$ Length Norm. | 44.27 | 0.75 | 42.37 | 1.45 | 61.37 | 58.94 | 1,942 |
| $\oplus$ Ref. Policy Mixing | 47.45 | 0.70 | 45.89 | 1.49 | 61.91 | 58.07 | 1890 |
| $\oplus$ Warm-up Adjustment | 48.52 | 0.80 | 45.88 | 1.45 | 63.37 | **59.95** | 1,919 |
| $\oplus$ Contextual Scaling | **51.66** | 0.78 | **47.92** | 1.49 | **63.94** | 59.69 | **1,878** |

Table 3: Evaluation of RAINBOWPO by adding new components consecutively.

**Ablations on RainbowPO.** We also conduct an ablation study of our proposed RAINBOWPO algorithm. All components of our proposed in our algorithm is useful, as in Table 4, for which we use $\oplus$ to denote that the methods are based on composition of the method on previous line and new elements in Table 4. We notice that adding length normalization is indeed important and of the most critical importance among the components for RainbowPO. We also include ablations on training epochs in Appendix E.3, which showcases that 3 epochs yield the sweet spot.

| Models | AlpacaEval2 (GPT4) | | | | AlpacaEval2 (Llama3-70B) | | |
|---|---|---|---|---|---|---|---|
| | LC WR (%) | $\sigma$ | WR (%) | $\sigma$ | LC WR (%) | WR (%) | avg length ($\downarrow$) |
| RainbowPO | **51.66** | 0.78 | **47.92** | 1.49 | 63.94 | 59.69 | 1,878 |
| $-$ Ref. Policy Mixing | 50.52 | 0.78 | 47.49 | 1.46 | 64.64 | 60.43 | 1,886 |
| $-$ Contextual Scaling | 48.52 | 0.80 | 45.88 | 1.45 | 63.37 | 59.95 | 1,919 |
| $-$ Length Normalization | 45.68 | 0.78 | 42.43 | 1.47 | 57.43 | 58.01 | 2108 |

Table 4: Ablation study of the newly proposed elements in RAINBOWPO.

## 4.2 COMPARISON WITH BASELINE METHODS

Table 6 shows the comparison between RainbowPO with the baselines. For a fair comparison, we first compare RainbowPO with the baselines in one training epoch, shown in Table 5. RainbowPO performs the best, beating SimPO while achieving lower average length.

| Models | AlpacaEval2 (GPT4) | | | AlpacaEval2 (Llama3-70B) | | |
|---|---|---|---|---|---|---|
| | LC WR (%) | WR (%) | $\sigma$ | LC WR (%) | WR (%) | avg length ($\downarrow$) |
| DPO (Rafailov et al., 2023) | 37.95 | 37.36 | 1.42 | 55.46 | 54.03 | 1,989 |
| IPO (Azar et al., 2024) | 34.80 | 34.52 | 1.40 | 52.67 | 50.93 | 1,956 |
| KTO (Ethayarajh et al., 2023) | 35.61 | 33.19 | 1.38 | 55.94 | 51.74 | 1,876 |
| CPO (Xu et al., 2024) | 31.89 | 34.92 | 1.38 | 53.33 | 54.84 | 2,155 |
| ORPO (Hong et al., 2024) | 22.91 | 22.59 | 1.24 | 48.41 | 45.90 | 1,914 |
| SimPO (Meng et al., 2024) | 47.96 | 41.17 | 1.44 | **61.94** | 54.22 | 1,730 |
| RainbowPO (1 epoch) | **48.08** | **42.53** | 1.43 | 61.36 | **54.60** | **1,683** |

Table 5: Methods comparison under one training epoch.

When we increased the training epoch to 3, interestingly, we also noticed that the same phenomenon as what (Meng et al., 2024) reported: SimPO rarely benefits from more epochs of training. However, RainbowPO and DPO both gets an increase in the winning rate after another two epochs of training, making the RainbowPO get a 51.66% win rate against GPT4 under GPT4 as a judge. This advantage not only benefits the final performance, but also might play larger impact when the alignment dataset is small or expensive to collect and will be beneficial to reuse, which is quite common in reality.

| Models | AlpacaEval2 (GPT4) | | | AlpacaEval2 (Llama3-70B) | | |
|---|---|---|---|---|---|---|
| | LC WR (%) | WR (%) | $\sigma$ | LC WR (%) | WR (%) | avg length ($\downarrow$) |
| DPO* (Rafailov et al., 2023) | 43.65 | 43.94 | 1.46 | 60.13 | 58.20 | 2,284 |
| SimPO* (Meng et al., 2024) | 48.40 | 44.57 | 1.47 | 60.90 | 56.46 | **1,843** |
| RainbowPO (3 epochs) | **51.66** | **47.92** | 1.49 | **63.94** | **59.69** | 1,878 |

Table 6: Methods comparison under three training epochs. *Hyper-parameters are further adjusted for the best performance.

### 4.3 LIMITATIONS AND FUTURE WORK

**Broader tasks.** In this paper, we focus our evaluation on models trained with LLama3-8B Instruct as the base model. Exploring other models of varying sizes, such as Gemma (Team et al., 2024) or Mistral (Jiang et al., 2023a), could possibly enhance the generalizability of our findings. It will also be beneficial if we could repeat the pipelines and compare the algorithms' performance on other LLM evaluation metrics, like arena-hard or MT-bench, though MT-bench is known to be less tinguishable for RLHF algorithms. Other directions include benchmarking the effectiveness of alignment algorithms on improving other capabilities of LLM other than instruction following, like reasoning (Xiong et al., 2024b) or COT (Choi et al., 2024). However, due to constraints in computing resources and time, we defer this investigation to future work. Nevertheless, we believe that our work provides a unified and comprehensive framework for helping to find the best preference optimization algorithms, and further pushing the boundary of offline RLHF for LLMs.

**Ideas from other xPOs.** We were not able to explore other aspects of existing DPO variants in detail, and there might be still promising candidates in further improving the preference of RainbowPO. Some methods that propose to update the reference policy dynamically: sDPO (Kim et al., 2024), TR-DPO (Gorbatovski et al., 2024); or learning from noisy preferences Chowdhury et al. (2024). Additionally, we also recognize the recent literature in pursuing online methods, such as online DPO (Guo et al., 2024) or iterative DPO (Yuan et al., 2024; Xiong et al., 2024a), which provide valuable insights on possibly further improving the downstream task performance: we will pursue them in future research. Other extensions beyond RLHF include, Nash Learning from human feedback (Munos et al., 2023), and self-play preference optimization (Chen et al., 2024b).

**Demystifying observations.** We also made some interesting observations in the paper, which we fail to find proper mathematical explanations and may boost further research. For example, the RainbowPO objective could benefit much more than SimPO when increasing the training epochs, but reasons for such phenomenons are still unknown. In addition, we found some mathematically orthogonal components are actually not empirically independent, for example, RSO can improve DPO, but can not be readily combined with other components like length normalization. It is also interesting to see the some combination of components reach effects "$1 + 1 > 2$"; it will be interesting to understand the deeper underlying reasons and could potentially lead to better algorithms.

## 5 CONCLUSION

In this paper, we propose RAINBOWPO, a comprehensive framework that demystifies and enhances existing DPO methods through the integration of key components into a unified objective. Our findings highlight the effectiveness of length normalization, reference policy mixing, and contextual scaling. However, the selective application of rejection sampling and home advantage is not providing incremental improvements either individually or when paired with the other methods. By demonstrating that these enhancements can coexist within a single algorithm to achieve state-of-the-art performance for fine-tuning SOTA commercial LLM, we pave the way for future research and practical applications. We aim for this work to serve as a foundation for refining DPO methodologies and to inspire further exploration of untested components for integrated agents.

## ACKNOWLEDGMENTS

We thank Stephen Rawls, Supriyo Chakraborty and Kartik Balasubramaniam for helpful feedbacks and pointers to relevant works. We also thank Benjamin Therien, Ashwinee Panda, Zain Sarwar, Akshaj Kumar Veldanda, and Andrei Mircea for helpful discussions.

Wenpin Tang and Hanyang Zhao are supported by NSF grant DMS-2206038, a start-up grant at Columbia University, and the Columbia Innovation Hub grant. The works of Hanyang Zhao and David D. Yao are part of a Columbia-CityU/HK collaborative project that is supported by InnoHK Initiative, The Government of the HKSAR and the AIFT Lab.

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

# A    BACKGROUND ON RLHF AND RL

## A.1    RLHF

RLHF (Ouyang et al., 2022; Stiennon et al., 2020; Ziegler et al., 2019). On top of $\pi^{\mathrm{SFT}}$, RLHF is proposed to serve as the next step to conduct further fine-tuning to generate high-quality outputs as judged by humans. Given a generative model $\pi$, the model $\pi$ is prompted with prompts $x$ to produce pairs of answers (or, "completions"), $\{y_1, y_2\} \sim \pi(y \mid x)$, which are then presented to human labelers who express preferences for one completion over the other. Denote by $y_w \succ y_l \mid x$, meaning that $y_w \in \{y_1, y_2\}$ is preferred over $y_l \in \{y_1, y_2\}$. The preferences are assumed to be generated by some latent reward model $r^*(x, y)$, which we do not have access to. Based on the collected preference data $\{x^{(i)}, y_w^{(i)}, y_l^{(i)}\}_{i=1}^N$, RLHF consists of first learning a reward model $r(x, y)$, followed by learning a policy $\pi_r(y \mid x)$ in which the prompt $x$ is the state, and the completion $y$ is the action.

(a) **Reward Model**. To capture the underlying human preferences, RLHF assumes the Bradley-Terry model (Bradley & Terry, 1952) that stipulates the pairwise preference distribution:

$$p^* (y_1 \succ y_2 \mid x) := \frac{\exp\left(r^*\left(x, y_1\right)\right)}{\exp\left(r^*\left(x, y_1\right)\right) + \exp\left(r^*\left(x, y_2\right)\right)} = \sigma\left(r^*\left(x, y_1\right) - r^*\left(x, y_2\right)\right), \quad (19)$$

where $\sigma(\cdot)$ is the sigmoid function. Given access to a static dataset of comparisons $\mathcal{D} = \{x^{(i)}, y_w^{(i)}, y_l^{(i)}\}_{i=1,\ldots,N}$, RLHF seeks to approximate the latent reward $r^*(x, y)$ by a family of functions $\{r_\psi(x, y)\}_\psi$, and estimate the parameters by minimizing the (negative) log-likelihood loss $\min_\psi \mathcal{L}(r_\psi, \mathcal{D}) := -\mathbb{E}_{(x, y_w, y_l) \sim \mathcal{D}} [\log \sigma (r_\psi(x, y_w) - r_\psi(x, y_l))]$. Denote by $r_{\psi_*}(x, y)$ the solution to this problem.

(b) **RL**. The learned reward function $r_{\psi_*}(x, y)$ is then used to provide feedback to the language model. More precisely, the following KL-regularized RL problem is considered:

$$\max_\pi \mathbb{E}_{x \sim \mathcal{D}} \left[\mathbb{E}_{y \sim \pi(y|x)} \left[r_{\psi_*}(x, y)\right] - \beta \mathrm{KL}\left(\pi(\cdot \mid x) \| \pi_{\mathrm{ref}}(\cdot \mid x)\right)\right], \quad (20)$$

where $\beta > 0$ is a hyper-parameter controlling the deviation from the reference policy $\pi_{\mathrm{ref}} = \pi^{\mathrm{SFT}}$. The regularization is important as it prevents deviating too far from the SFT model that is trained to conform to the true preference, while maintaining the generation diversity to avoid mode-collapsing to a single high-reward answer. In view of Equation 20, RLHF uses the reward function $r(x, y) = r_\psi(x, y) - \beta(\log \pi(y \mid x) - \log \pi_{\mathrm{ref}}(y \mid x))$, and solves the RL problem by proximal policy optimization (PPO) (Schulman et al., 2017).

## A.2    DPO

One disadvantage of RLHF is that the RL step often requires substantial computational effort (e.g., to carry out PPO). The idea of DPO is to combine the reward model and RL in RLHF into a single objective, bypassing the computation in the RL step. The key realization is that given a reward function $r(x, y)$, the RL problem in Equation 20 has a closed-form solution $\pi_r(y \mid x) = \frac{1}{Z(x)} \pi_{\mathrm{ref}}(y \mid x) \exp\left(\frac{1}{\beta} r(x, y)\right)$, where $Z(x) = \sum_y \pi_{\mathrm{ref}}(y \mid x) \exp\left(\frac{1}{\beta} r(x, y)\right)$. Rewrite the above as $r(x, y) = \beta \log \frac{\pi_r(y|x)}{\pi_{\mathrm{ref}}(y|x)} + \beta \log Z(x)$. Through this change of variables, the latent reward $r^*(x, y)$ can be expressed in terms of the optimal policy $\pi^*(y \mid x)$, the reference policy $\pi_{\mathrm{ref}}(y \mid x)$ and a constant $Z^*(x)$. Substituting this $r^*$ expression into Equation 19 yields:

$$p^*(y_1 \succ y_2 \mid x) = \sigma\left(\beta \log \frac{\pi^*(y_1 \mid x)}{\pi_{\mathrm{ref}}(y_1 \mid x)} - \beta \log \frac{\pi^*(y_2 \mid x)}{\pi_{\mathrm{ref}}(y_2 \mid x)}\right), \quad (21)$$

where $Z^*(x)$ cancels out. the preference distribution only depends on $\pi^*(y \mid x)$ and $\pi_{\mathrm{ref}}(y \mid x)$. The expression in Equation 21 motivates the DPO objective:

$$\min_\theta \mathcal{L}_{\mathrm{DPO}}(\pi_\theta; \pi_{\mathrm{ref}}) := -\mathbb{E}_{(x, y_w, y_l) \sim \mathcal{D}} \left[\log \sigma\left(\beta \log \frac{\pi_\theta(y_w \mid x)}{\pi_{\mathrm{ref}}(y_w \mid x)} - \beta \log \frac{\pi_\theta(y_l \mid x)}{\pi_{\mathrm{ref}}(y_l \mid x)}\right)\right],$$
$$(22)$$

# B  MISCELLANEOUS

Table 7 lists the common notations used in this paper. The table serves as a quick reference guide for understanding the mathematical expressions and technical terms used throughout the paper.

| Name | Notation | Description |
|---|---|---|
| Input Sequence | $x$ | Input sequence that is passed to the model. |
| Output Sequence | $y$ | Expected label or output of the model. |
| Dispreferred Response | $y_l$ | Negative samples for reward model training. |
| Preferred Response | $y_w$ | Positive samples for reward model training. |
| Optimal Policy Model | $\pi^*$ | Optimal policy model. |
| Policy Model | $\pi_\theta$ | Generative model that takes the input prompt and returns a sequence of output or probability distribution. |
| Reference Policy Model | $\pi_{\mathrm{ref}}$ | Generative model that is used as a reference to ensure the policy model is not deviated significantly. |
| Preference Dataset | $\mathcal{D}$ | Dataset with a set of preferred and dispreferred responses. |
| Preference Dataset by RSO | $\mathcal{D}_{\mathrm{RSO}}$ | Dataset with a set of preferred and dispreferred responses sampled by Rejection Sampling. |
| Loss Function | $\mathcal{L}$ | Loss function. |
| Regularization Hyper-parameter | $\beta$ | Regularization Hyper-parameter for preference tuning. |
| Mixing Hyper-parameter | $\alpha$ | Our proposed mixing coefficient for better reference policy. |
| Reward | $r$ | Reward score. |
| Target Reward Margin | $\gamma$ | The margin separating the winning and losing responses. |

Table 7: Table of Terminology and Notation.

## C  MATHEMATICAL EXPLANATION OF DIFFERENT XPOS

Here we first list popular XPOs variants in the literature as in Meng et al. (2024) and Winata et al. (2024).

| Method | Objective |
|---|---|
| DPO | $-\log \sigma \left( \beta \log \frac{\pi_\theta(y_w|x)}{\pi_{\text{ref}}(y_w|x)} - \beta \log \frac{\pi_\theta(y_l|x)}{\pi_{\text{ref}}(y_l|x)} \right)$ |
| IPO | $\left( \beta \log \frac{\pi_\theta(y_w|x)}{\pi_{\text{ref}}(y_w|x)} - \beta \log \frac{\pi_\theta(y_l|x)}{\pi_{\text{ref}}(y_l|x)} - \frac{1}{2} \right)^2$ |
| $f$-DPO | $-\log \sigma \left( \beta f' \left( \frac{\pi_\theta(y_w|x)}{\pi_{\text{ref}}(y_w|x)} \right) - \beta f' \left( \frac{\pi_\theta(y_l|x)}{\pi_{\text{ref}}(y_l|x)} \right) \right)$ |
| KTO | $-\lambda_w \sigma \left( \beta \log \frac{\pi_\theta(y_w|x)}{\pi_{\text{ref}}(y_w|x)} - z_{\text{ref}} \right) - \lambda_l \sigma \left( z_{\text{ref}} - \beta \log \frac{\pi_\theta(y_l|x)}{\pi_{\text{ref}}(y_l|x)} \right)$, 
 where $z_{\text{ref}} = \mathbb{E}_{(x,y)\sim\mathcal{D}} \left[ \beta \text{KL} \left( \pi_\theta(y|x) || \pi_{\text{ref}}(y|x) \right) \right]$ |
| ODPO | $-\log \sigma \left( \beta \log \frac{\pi_\theta(y_w|x)}{\pi_{\text{ref}}(y_w|x)} - \beta \log \frac{\pi_\theta(y_l|x)}{\pi_{\text{ref}}(y_l|x)} - \Delta_r(x) \right)$ |
| MallowsPO | $-\log \sigma \left( \phi(x) \left[ \beta \log \frac{\pi_\theta(y_w|x)}{\pi_{\text{ref}}(y_w|x)} - \beta \log \frac{\pi_\theta(y_l|x)}{\pi_{\text{ref}}(y_l|x)} \right] \right)$ |
| R-DPO | $-\log \sigma \left( \beta \log \frac{\pi_\theta(y_w|x)}{\pi_{\text{ref}}(y_w|x)} - \beta \log \frac{\pi_\theta(y_l|x)}{\pi_{\text{ref}}(y_l|x)} - (\alpha|y_w| - \alpha|y_l|) \right)$ |
| CPO | $-\log p_\theta(y_w|x) - \log \sigma \left( \beta \log \pi_\theta(y_w|x) - \beta \log \pi_\theta(y_l|x) \right)$ |
| ORPO | $-\log p_\theta(y_w|x) - \lambda \log \sigma \left( \log \frac{p_\theta(y_w|x)}{1 - p_\theta(y_w|x)} - \log \frac{p_\theta(y_l|x)}{1 - p_\theta(y_l|x)} \right)$, 
 where $p_\theta(y|x) = \exp \left( \frac{1}{|y|} \log \pi_\theta(y|x) \right)$ |
| SimPO | $-\log \sigma \left( \frac{\beta}{|y_w|} \log \pi_\theta(y_w|x) - \frac{\beta}{|y_l|} \log \pi_\theta(y_l|x) - \gamma \right)$ |

Table 8: Various preference optimization DPO objectives. The table is inspired from Meng et al. (2024) and Winata et al. (2024).

Next we include the categorization of different methods, which appeared earlier in Table 9, and explain detailedly the reasons after.

| Method | Length Norm. | Link Func. | Home Adv. | Ref. Policy | Contextual Scaling | RS | SFT Loss |
|---|---|---|---|---|---|---|---|
| DPO | × | logistic | × | SFT | × | × | × |
| SLiC-HF | × | hinge | × | SFT | × | × | ✓ |
| IPO | × | square | × | SFT | × | × | × |
| CPO | × | logistic | × | Free | × | × | ✓ |
| RSO | × | logistic / hinge | × | SFT | × | ✓ | × |
| ODPO | × | logistic | ✓ | SFT | × | × | × |
| ORPO | ✓ | logistic | × | Free | implicitly | × | ✓ |
| WPO | × | logistic | × | SFT | ✓ | × | × |
| MallowsPO | × | logistic | × | SFT | ✓ | × | × |
| SimPO | ✓ | logistic | ✓ | Free | × | × | × |
| RainbowPO | ✓ | logistic | × | mixing | ✓ | × | × |

Table 9: Mapping of XPOs with mathematically orthogonal components and validation results of their effectiveness by the downstream task evaluations.

## C.1 DIRECT PREFERENCE OPTIMIZATION (DPO)

The loss of DPO (Rafailov et al., 2023) is:

$$\mathcal{L}_{\text{DPO}}\left(\pi_{\theta}; \pi_{\text{ref}}\right) := -\mathbb{E}_{(x,y_w,y_l)\sim\mathcal{D}}\left[\log\sigma\left(\beta\log\frac{\pi_{\theta}\left(y_w \mid x\right)}{\pi_{\text{ref}}\left(y_w \mid x\right)} - \beta\log\frac{\pi_{\theta}\left(y_l \mid x\right)}{\pi_{\text{ref}}\left(y_l \mid x\right)}\right)\right], \quad (23)$$

which can be seen that, as the baseline methods we investigate in this paper, no length normalization is adopted, link function $-\log\sigma$ adopts the logistic function, home advantage can be seen as None or 0, reference policy is the SFT policy to be aligned, no contextual scaling, the dataset is the hybrid or offline dataset thus there is no rejection sampling. Also there is no SFT loss.

## C.2 SEQUENCE LIKELIHOOD CALIBRATION FROM HUMAN FEEDBACK (SLiC-HF)

The loss of SliC-HF (Zhao et al., 2023) is ($y_{\text{ref}}$ refers to answer generated by $\pi_{\text{ref}}$):

$$\mathcal{L}_{\text{SLiC}}(\pi_{\theta}; \pi_{\text{ref}}) = \mathbb{E}_{(x,y_w,y_l)}\underbrace{\max\left(0, \delta - \log\pi_{\theta}(y_w|x) + \log\pi_{\theta}(y_l|x)\right)}_{\text{rank calibration loss}}\underbrace{-\lambda\log\pi_{\theta}(y_{\text{ref}}|x)}_{\text{SFT}}, \quad (24)$$

which can be seen that, no length normalization is adopted, link function $\max(0, \delta - x)$ adopts the hinge function, home advantage can be seen as None or 0, reference policy is the SFT policy to be aligned (in the regularization term), no contextual scaling, the dataset is the hybrid or offline dataset thus there is no rejection sampling. There is an SFT loss, which acts as the role of regularization.

## C.3 IDENTITY PREFERENCE OPTIMIZATION (IPO)

The loss of IPO (Azar et al., 2024) is:

$$\mathcal{L}_{\text{IPO}}\left(\pi_{\theta}; \pi_{\text{ref}}\right) := \mathbb{E}_{(x,y_w,y_l)\sim\mathcal{D}}\left(\beta\log\frac{\pi_{\theta}\left(y_w \mid x\right)}{\pi_{\text{ref}}\left(y_w \mid x\right)} - \beta\log\frac{\pi_{\theta}\left(y_l \mid x\right)}{\pi_{\text{ref}}\left(y_l \mid x\right)} - \frac{1}{2}\right)^2, \quad (25)$$

which can be seen that, no length normalization is adopted, link function $(x - 1/2)^2$ adopts the square function, home advantage can be seen as None or 0, reference policy is the SFT policy to be aligned, no contextual scaling, the dataset is the hybrid or offline dataset thus there is no rejection sampling. Also there is no SFT loss.

## C.4 CPO

CPO (Xu et al., 2024) is motivated to improve the memory and speed efficiency of DPO by neglecting the reference policy, further accompanied by a SFT loss term:

$$\mathcal{L}_{\text{CPO}}\left(\pi_{\theta}\right) := -\mathbb{E}_{(x,y_w,y_l)\sim\mathcal{D}}\left[\log p_{\theta}(y_w|x) + \log\sigma\left(\beta\log\frac{\pi_{\theta}(y_w|x)}{\pi_{\theta}(y_l|x)}\right)\right]. \quad (26)$$

which can be seen that, no length normalization is adopted, link function $-\log\sigma$ adopts the logistic function, home advantage can be seen as None or 0, reference policy is None (or free), no contextual scaling, the dataset is the hybrid or offline dataset thus there is no rejection sampling. Also there is an SFT loss.

## C.5 RSO

The loss of RSO (Rafailov et al., 2023) is:

$$\mathcal{L}_{\text{RSO}}\left(\pi_{\theta}; \pi_{\text{ref}}\right) := -\mathbb{E}_{(x,y_w,y_l)\sim\mathcal{D}_{\text{RS}}}\left[\log\sigma\left(\beta\log\frac{\pi_{\theta}\left(y_w \mid x\right)}{\pi_{\text{ref}}\left(y_w \mid x\right)} - \beta\log\frac{\pi_{\theta}\left(y_l \mid x\right)}{\pi_{\text{ref}}\left(y_l \mid x\right)}\right)\right], \quad (27)$$

which can be seen that, it only differs from DPO in applying rejection sampling for formulating the preference dataset.

## C.6 ODPO

ODPO (Amini et al., 2024) proposed to add a margin to capture the significance of preference pairs; they model this margin, or they call offset $\Delta_r$ as a monotonically increasing function $f(\cdot)$ of the difference between the scores associated with the responses:

$$\Delta_r(x, y_w, y_l) = \alpha f\left(\text{score}\,(x, y_w) - \text{score}\,(x, y_l)\right),\tag{28}$$

where $\alpha$ is a hyper-parameter that controls the extent to which an offset should be enforced. The resulting objective becomes:

$$\mathcal{L}_{\text{ODPO}}\left(\pi_\theta; \pi_{\text{ref}}\right) :=$$
$$-\,\mathbb{E}_{(x, y_w, y_l) \sim \mathcal{D}}\left[\log \sigma\left(\beta \log \frac{\pi_\theta(y_w|x)}{\pi_{\text{ref}}(y_w|x)} - \beta \log \frac{\pi_\theta(y_l|x)}{\pi_{\text{ref}}(y_l|x)} - \Delta_r(x, y_w, y_l)\right)\right].\tag{29}$$

ODPO only differs from DPO in applying a contextual dependent margin/home advantage.

## C.7 ORPO

Opposed to maximizing the likelihood ratios of winning and losing answers in the preference pair in DPO, ORPO (Hong et al., 2024) propose that odds ratio can be a more sensible choice.

$$\mathcal{L}_{\text{ORPO}}\left(\pi_\theta\right) :=$$
$$-\,\mathbb{E}_{(x, y_w, y_l) \sim \mathcal{D}}\left[\log p_\theta(y_w|x) + \lambda \log \sigma\left(\log \frac{p_\theta(y_w|x)}{1 - p_\theta(y_w|x)} - \log \frac{p_\theta(y_l|x)}{1 - p_\theta(y_l|x)}\right)\right].\tag{30}$$

where $p_\theta(y|x) = \exp\left(\frac{1}{|y|}\log \pi_\theta(y|x)\right)$. ORPO is similar to CPO in the sense that it is also reference model free and combined with a SFT loss; in addition, notably that ORPO also adopts a form of length regularization by normalizing the likelihoods with respect to the length, as in the definition of $p_\theta(y|x)$; finally, they compute odds ratio instead of the original likelihood ratio.

ORPO differs from DPO in being reference-model free and also yields a SFT term for regularization. Its contextual scaling is implicit, as shown in Theorem 1 for ORPO equivalent objective.

## C.8 MALLOWSPO

Chen et al. (2024a) propose a contextual scaled objective derived from MLE under Mallows: compared to DPO that puts equal weights on each prompt and preference pairs, the resulting MallowsPO adds a contextual scaling factor $\phi(x)$ that represents this dispersion of the preferences of answers to each prompt $x$:

$$\mathcal{L}_{\text{MallowsPO}}\left(\pi_\theta; \pi_{\text{ref}}\right) :=$$
$$-\,\mathbb{E}_{(x, y_w, y_l) \sim \mathcal{D}}\left[\log \sigma\left(\phi(x)\left[\beta \log \frac{\pi_\theta(y_w|x)}{\pi_{\text{ref}}(y_w|x)} - \beta \log \frac{\pi_\theta(y_l|x)}{\pi_{\text{ref}}(y_l|x)}\right]\right)\right].\tag{31}$$

To compute this dispersion, MallowsPO provided a direct approach by using a normalized predictive entropy of preference pairs $\{y_i^w, y_i^l\}_{i=1,\dots,N}$ with $N = \max(|y^w|, |y^l|)$:

$$\phi(x) = -\log\left(\frac{\frac{1}{2}\sum_{i=1}^{N-1}\left[H_{\pi_{\text{ref}}}(Y_{i+1} \mid Y_i = y_i^w) + H_{\pi_{\text{ref}}}(Y_{i+1} \mid Y_i = y_i^l)\right]}{\log(n)}\right).\tag{32}$$

MallowsPO differs from DPO in adding a contextual scaling factor; others are the same.

## C.9 SIMPO

SimPO (Meng et al., 2024) proposes a simple yet effective objective that is claimed to match or even outperform the performance of DPO:

$$\mathcal{L}_{\text{SimPO}}\left(\pi_\theta\right) := -\mathbb{E}_{(x, y_w, y_l) \sim \mathcal{D}}\left[\log \sigma\left(\frac{\beta}{|y_w|}\log \pi_\theta(y_w|x) - \frac{\beta}{|y_l|}\log \pi_\theta(y_l|x) - \gamma\right)\right],\tag{33}$$

where $\gamma$ is introduced as a target reward margin to help separating the winning and losing responses. SimPO is similar to CPO in the sense of being reference model free; it also adopted the length normalization for the likelihoods as in ORPO; finally, it additionally introduced a constant margin to be tuned that could help to further improve the performance by encouraging a larger difference between the normalized likelihoods.

SimPO differs from DPO in adopting length normalization, a margin/home advantage term and it is also reference-model free.

## D  PROOFS AND DETAILS

### D.1  PROOF OF ORPO UPPER BOUND IN EQUATION 4

Here we prove that the part of preference optimization in ORPO's loss yields an upper bound which has instinct connection to SimPO loss, specifically the idea of length normalization.

**Theorem 1.** *Assume that the normalized implicit reward model difference for preference pairs:*

$$\Delta_\theta(x, y_w, y_l) = \frac{1}{|y_w|} \log \pi_\theta(y_w|x) - \frac{1}{|y_l|} \log \pi_\theta(y_l|x) \geq 0$$

*almost surely. Then for the part of preference optimization in ORPO loss, i.e.*

$$\mathcal{L}_{\text{ORPO-PO}}(\pi_\theta) = -\mathbb{E}_{(x,y_w,y_l)\sim\mathcal{D}} \left[ \log \sigma \left( \log \frac{p_\theta(y_w|x)}{1 - p_\theta(y_w|x)} - \log \frac{p_\theta(y_l|x)}{1 - p_\theta(y_l|x)} \right) \right], \qquad (34)$$

*has an upper bound such that*

$$\mathcal{L}_{\text{ORPO-PO}} \leq -\mathbb{E} \log \sigma \left( \frac{1}{1 - p_\theta(y_l|x)} \left( \frac{1}{|y_w|} \log \pi_\theta(y_w|x) - \frac{1}{|y_l|} \log \pi_\theta(y_l|x) \right) \right). \qquad (35)$$

*Proof.* Since $-\log \sigma(\cdot)$ is a monotone decreasing function, when $1 > x > y > 0$, it suffices to prove that for any $x$,

$$\log \left( \frac{x}{1-x} \right) - \log \left( \frac{y}{1-y} \right) \geq \frac{1}{1-y} \log \left( \frac{x}{y} \right), \qquad (36)$$

in which $x = p_\theta(y_w|x)$, $y = p_\theta(y_l|x)$. The inequality is equivalent to:

$$f(x) := \log \left( \frac{x}{1-x} \right) - \frac{1}{1-y} \log \left( \frac{x}{y} \right) \geq \log \left( \frac{y}{1-y} \right). \qquad (37)$$

Taking gradient of $f(x)$ with respect to $x$, we have:

$$f'(x) = \frac{1}{x} + \frac{1}{1-x} - \frac{1}{1-y} \cdot \frac{1}{x} = \frac{1}{x(1-x)} - \frac{1}{x(1-y)} \geq 0.$$

Moreover, we have $f(y) = \log(\frac{y}{1-y})$, which yields the desired result. $\qquad \square$

# E    EXPERIMENTAL DETAILS

## E.1    ALPACAEVAL AND ALPACAEVAL2 INTRODUCTION

We briefly introduce AlpacaEval (Li et al., 2023) and its evaluation pipeline and metric. This part is mainly modified and summarized from descriptions in AlpacaEval official Github repository.[5]

In practice, for either training a LLM or comparison between different LLMs, we need to evaluate an instruction-following model (e.g., ChatGPT). However, evaluation of such models typically requires human interactions, which is time-consuming, expensive, and hard to replicate.

**AlpacaEval**. AlpacaEval is an LLM-based automatic evaluation that is fast, cheap, replicable, and validated against 20K human annotations. It operates on a fixed set of 805 instructions chosen to be representative of user interactions on the Alpaca web demo,[6] and has been widely adopted for model development. Concretely, AlpacaEval adopts an automatic evaluator that has high agreement with humans (validated on 20K annotations), and evaluates a model by measuring the fraction of times a powerful LLM (e.g., GPT-4) prefers the outputs from that model over outputs from a reference model. The evaluators of AlpacaEval enable caching and output randomization by default.

**AlpacaEval2**. AlpacaEval2 (Dubois et al., 2024), also called Length Controlled AlpacaEval, is a length-debiased version of AlpacaEval. One of the major issues of AlpacaEval is that one can increase the win-rate by increasing the length of outputs. The main idea of Length Controlled (LC) WR is that for each model, AlpacaEval2 will fit a logistic regression to predict the preference of the autoannotator given: (1) the instruction, (2) the model, and (3) the difference of length between the baseline and model output. Given such a logistic regression AlpacaEval2 can then try to predict the counterfactual "what would the preference be if the model's output had the same length as the baseline" by setting the length difference to 0. By averaging over this length-controlled preference, AlpacaEval2 then obtains the LC win rates.

Length controlled win-rates increase the correlation between AlpacaEval's leaderboard and Chat Arena from 0.93 to 0.98 Spearman correlation, while significantly decreasing the length gameability of the annotator. We refer the more concrete details of the datasets, comparison of models or evaluators and leaderboards to AlpacaEval's official website `https://github.com/tatsu-lab/stanford_alpaca`.

**Our paper**. In this paper, we adopted two evaluators: `GPT4 turbo` and Llama3 70B to evaluate our models' generations against base/reference answers generated by `GPT4 turbo`. We adopt the default template for evaluators provided by AlpacaEval and we adopt the following fixed generation config for each model: max new tokens 4096, temperature 0.7 and top $p$ 0.1.

## E.2    TRAINING DETAILS

Here we report the best hyper-parameters we searched which corresponds to our final results. We include the modified dpo trainer and training scripts in the supplementary materials.

## E.3    ABLATIONS ON TRAINING EPOCHS

We also conduct ablation studies on the training epochs, under the other best hyper-parameters we found under the 3 training epochs. The training loss can be found in Figure 2, and the AlpacaEval results could be found in Table 13. The results indeed show that 3 training epochs yield the sweet spot, further additional epochs could slightly degrade the performance.

---

[5]`https://github.com/tatsu-lab/alpaca_eval`.
[6]`https://github.com/tatsu-lab/stanford_alpaca`.

| Models | $\beta$ | $\alpha$ | $\gamma$ | $\tau$ | SFT $\lambda$ | lr | WR |
|---|---|---|---|---|---|---|---|
| Base model | 0.01 | 1 | 0 | $\infty$ | 0 | $3e^{-7}$ | 0.1 |
| + Length Norm. (LN) | 10 | 1 | 0 | $\infty$ | 0 | $e^{-6}$ | 0.1 |
| + Ref. Policy Mixing (Mix) | 0.01 | 0.25 | 0.1 | $\infty$ | 0 | $3e^{-7}$ | 0.1 |
| + Contextual Scaling (CS) | 0.01 | 1 | 0 | $\infty$ | 0 | $3e^{-7}$ | 0.1 |
| + Link Function (LF) | 0.001 | 1 | 0 | $\infty$ | 0 | $3e^{-7}$ | 0.1 |
| + Home Advantage (HA) | 0.005 | 1 | 0.001 | $\infty$ | 0 | $3e^{-7}$ | 0.1 |
| + Rejection Sampling (RSO) | 0.01 | 1 | 0 | 0.2 | 0 | $3e^{-7}$ | 0.1 |
| Base model + LN | 10 | 1 | 0 | $\infty$ | 0 | $e^{-6}$ | 0.1 |
| + LN + Mix | 10 | 0.25 | 0.1 | $\infty$ | 0 | $e^{-6}$ | 0.1 |
| + LN + CS | 10 | 1 | 0 | $\infty$ | 0 | $e^{-6}$ | 0.1 |
| + LN + HA | 10 | 1 | 0.05 | $\infty$ | 0 | $e^{-6}$ | 0.1 |
| + LN + RS | 10 | 1 | 0 | 0.2 | 0 | $e^{-6}$ | 0.1 |
| + LN + SFT Loss | 10 | 1 | 0 | $\infty$ | 0.1 | $e^{-6}$ | 0.1 |

Table 10: Hyper-parameters for results reported in Table 2.

| Models | $\beta$ | $\alpha$ | $\gamma$ | lr | WR/WS |
|---|---|---|---|---|---|
| Base model | 0.01 | 1 | 0 | $3e^{-7}$ | 0.1 |
| + Length Norm. (LN) | 10 | 1 | 0 | $e^{-6}$ | 0.1 |
| + Ref. Policy Mixing (Mix) | 10 | 0.25 | 0.1 | $e^{-6}$ | 0.1 |
| + Warm-up Adjustment | 10 | 0.25 | 0.1 | $e^{-6}$ | 150 |
| + Contextual Scaling (CS) | 10 | 0.25 | 0.1 | $e^{-6}$ | 150 |

Table 11: Hyper-parameters for results reported in Table 3.

| Models | $\beta$ | $\alpha$ | $\gamma$ | lr | WR/WS |
|---|---|---|---|---|---|
| DPO* (Rafailov et al., 2023) | 0.01 | 1 | 0 | $3e^{-7}$ | 150 |
| SimPO* (Meng et al., 2024) | 10 | 0 | 0.1 | $e^{-6}$ | 150 |
| RainbowPO* (3 epochs) | 10 | 0.25 | 0.1 | $e^{-6}$ | 150 |

Table 12: Hyper-parameters for Table 6.

| Models | AlpacaEval2 (GPT4) | | | | |
|---|---|---|---|---|---|
| | LC WR (%) | $\sigma$ | WR (%) | $\sigma$ | avg length ($\downarrow$) |
| RainbowPO | **51.66** | 0.78 | **47.92** | 1.49 | 1,878 |
| 1 Training Epoch | 46.36 | 0.77 | 39.32 | 1.43 | 1,717 |
| 2 Training Epochs | 50.88 | 0.77 | 47.34 | 1.47 | 1,912 |
| 4 Training Epochs | 50.61 | 0.74 | 47.02 | 1.49 | 1874 |

Table 13: Ablation of training epochs for RAINBOWPO (3 training epochs) using the same other hyper-parameters.

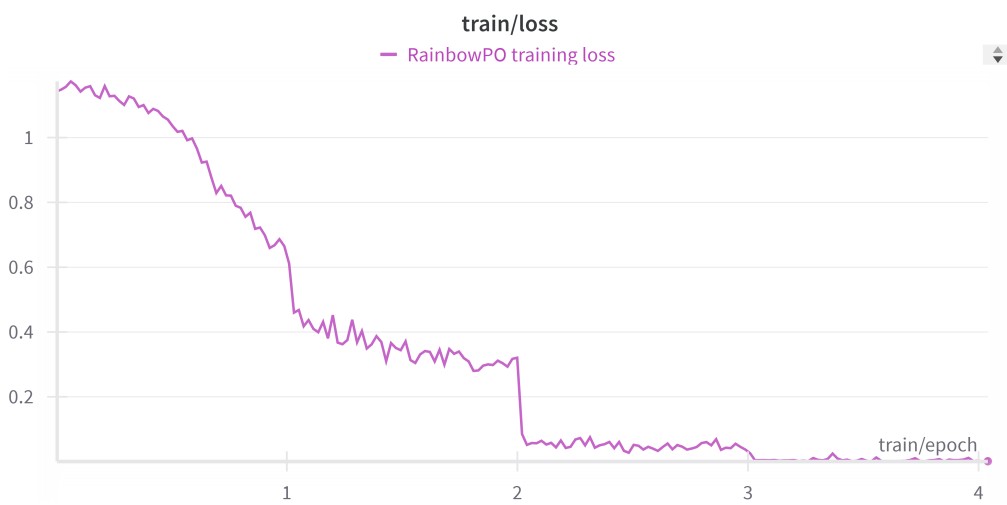

Figure 2: Training loss with respect to training epochs.

