# OpenReview forum: "RainbowPO: A Unified Framework for Combining Improvements in Preference Optimization"
_ICLR.cc/2025/Conference — ICLR 2025 Poster_

### Official Review · Reviewer_89tQ · 2024-11-03

**Soundness:** 3
**Presentation:** 3
**Contribution:** 3
**Rating:** 6
**Confidence:** 4

**Summary:**

Direct Preference Optimization (DPO) has emerged as an effective approach for fine-tuning large language models (LLMs) to align with human preferences. Various versions of DPO have been proposed in the literature; however, an extensive comparison between these different DPO variants is still lacking.
This paper aims to analyze various versions of DPO to identify components that enhance performance across different applications. To this end,  seven key components have been identified  across ten DPO variants. By combining these components, a new approach that achieves state-of-the-art performance has been proposed. This approach is conceptually similar to the Rainbow framework [1], which unified various enhancements of Deep Q-Networks (DQNs) from the literature.

This paper explains seven components, namely--Length Normalization, Link Function, Home Advantage, Mixing Reference Policy, Rejection Sampling, Contextual Scaling and SFT Loss. Then a new loss is formulated combining the seven components. Finally, the experimental evaluation reveals that Length Normalization, Contextual Scaling  and Rejection Sampling contribute to improvement over the baseline DPO.

The paper could be a significant contribution in the field of Fine-tuning large language models.
However, to meet the publication standards of a top-tier conference, it requires an additional round of thorough editing.

[1] Hessel, Matteo, et al. "Rainbow: Combining improvements in deep reinforcement learning." Proceedings of the AAAI conference on artificial intelligence. Vol. 32. No. 1. 2018.

**Strengths:**

**Originality:** This paper performs a meta-analysis on different variants of DPO.

**Quality**: The experimental evaluation comparing different components is the main strength of the paper.

**Clarity**: The text is written with clarity and well organized. It is easy to follow, even for someone like me, who is not up-to-date with the recent developments in LLM finetuning techniques. However, some of the terminologies introduced require explanations (see weakness).

**Significance**: Fine-tuning large language models (LLMs) for preference optimization has garnered increasing attention recently. A meta-study comparing different variants of preference optimization in LLMs would be highly valuable for both academics and practitioners. This work combines seven variants and promises to make the implementation publicly available, which would be a significant contribution to the community.

**Weaknesses:**

- Some acronyms, such as IPO and KTO, are never explained. Given that this paper is a meta-analysis, it’s important to define all acronyms for the reader’s convenience.

- Define LC WR in the caption or in a footnote in Table 4.

- It seems the term 'warm-up adjustment'  is never introduced before Table 3.

- The color coding in Table 4 is counterintuitive; using red to indicate better-performing models and green for worse-performing models is disorienting for the readers.

- The results in Table 4 are based on training for 3 epochs. Could these outcomes differ if the model were trained for additional epochs? Maybe you can  conduct an ablation study on the impact of number of epochs or plot training loss against number of epochs.

- Is there any explanation why rejection sampling does not perform well with length normalization?

I would be happy to raise my score, if the authors reply to my questions.

**Questions:**

- DPO techniques have been reported to lack generalizability in out-of-distribution tasks [1]. To address this limitation, learning chain-of-thought (CoT) policies has been proposed in the literature. Can the method proposed in this paper be adapted to learn a CoT policy?

- Robust DPO [2] is another approach to make DPO robust to noisy preferences. Is there a reason why this variant is not considered in your work?

- Have you considered approaches [3] which removes the process of learning a reward model by directly learning though supervised learning?

- Apart from Win ratio (WR), have you measured BERTScore?

1: Choi, Eugene, et al. "Robust Chain of Thoughts Preference Optimization." Seventeenth European Workshop on Reinforcement Learning.

2: Chowdhury, Sayak Ray, Anush Kini, and Nagarajan Natarajan. "Provably Robust DPO: Aligning Language Models with Noisy Feedback."
 Forty-first International Conference on Machine Learning.

3: Hejna, Joey, et al. "Contrastive Preference Learning: Learning from Human Feedback without Reinforcement Learning." The Twelfth International Conference on Learning Representations.

---

> ### Author Response · Authors · 2024-11-18
> **Revised Writing for Enhanced Clarity and Addressing Reviewer Feedback**
>
> Thank you for your detailed review and feedback. We greatly appreciate your recognition of the significance of our work and its potential impact on the field and future research. Below, we have addressed your questions as follows:
>
> > Some acronyms, such as IPO and KTO, are never explained. Given that this paper is a meta-analysis, it’s important to define all acronyms for the reader’s convenience.
>
> Thank you for highlighting this issue. We have revised the second paragraph of **Section 1** to ensure that every acronym used in the paper is clearly defined.
>
> > Define LC WR in the caption or in a footnote in Table 4.
>
> We have included additional details about this abbreviation in the **Experimental Setup** section, specifically addressing the Length Controlled (LC) Win Rate (WR) logic. These updates have been highlighted in blue.
>
> > It seems the term 'warm-up adjustment' is never introduced before Table 3.
>
> Thank you for bringing this to our attention. Due to space constraints, we have introduced "warm-up adjustment" and provided a detailed description of it in the appendix. Please refer to **Appendix E.2** for comprehensive information about the training process.
>
> > The color coding in Table 4 is counterintuitive; using red to indicate better-performing models and green for worse-performing models is disorienting for the readers.
>
> We have updated the coloring scheme for **Table 4** by reversing it.
>
> > The results in Table 4 are based on training for 3 epochs. Could these outcomes differ if the model were trained for additional epochs? Maybe you can conduct an ablation study on the impact of the number of epochs or plot training loss against the number of epochs.
>
> Thank you for your suggestion! We have retrained our RainbowPO model for 4 epochs and included a figure of the training loss curve relative to the training epochs in **Appendix E.3**. Additionally, we evaluated the models using AlpacaEval, which demonstrates that *training for 3 epochs yields the best results, outperforming training for either 2 or 4 epochs*. Please refer to **Appendix E.3** for detailed statistics.
>
> > Is there any explanation why rejection sampling does not perform well with length normalization?
>
> Currently, we have not identified a theoretical explanation to support our empirical findings. Our results indicate that rejection sampling and length normalization cannot simply be combined to complement each other effectively. We intend to investigate this phenomenon further to gain a deeper theoretical understanding in the future.
>
> > Can the method proposed in this paper be adapted to learn a CoT policy?
>
> Thank you for bringing this paper to our attention. We believe that RainbowPO has the potential to enhance the policy trained by the DPO and IPO methods outlined in this paper, leading to improved performance. This would further highlight the potential and significance of our work. We plan to explore this direction in future research and have now referenced this paper in **Section 4.3**, which discusses other DPO approaches. We mark the change in blue.
>
> > Robust DPO [2] is another approach to make DPO robust to noisy preferences. Is there a reason why this variant is not considered in your work?
>
> Thank you for bringing this paper to our attention. While we did not initially consider the direction of noisy preferences, we believe it represents a promising avenue that could benefit RainbowPO as well. We have added this paper to **Section 4.3**, which discusses other DPOs. We acknowledge that there are other potentially beneficial directions, but due to a limited computational budget, we were unable to explore them all. We plan to pursue these in future research. The changes have been highlighted in blue.
>
> > Have you considered approaches [3] which removes the process of learning a reward model by directly learning through supervised learning?
>
> For the CPO loss objective, it is actually a special instance of our RainbowPO objective by removing the reference policy, removing contextual scaling and adding back the SFT loss; RainbowPO also does not contain learning a reward model, and we align our model directly through supervised learning as well.
>
> >Apart from Win ratio (WR), have you measured BERTScore?
>
> We didn’t compute BERTScore because AlpacaEval does not have a golden (or reference) answer to such computations. We included more details about AlpacaEval in both Experiment Setup in **Section 4** and **Appendix E.1**

---

> > ### Comment · Reviewer_89tQ · 2024-11-21
> > **Response to Rebuttal**
> >
> > Thank you very much for addressing my questions and revising the paper. While I haven’t gone through it in detail yet, at first glance, the updated version appears much more polished.
> >
> > I understood that learning reward models is not part of RainbowPO's training. I was not sure whether Contrastive Preference Learning can be implemented with RainbowPO. I appreciate your response on that.
> >
> > Based on the revised version I will raise  my score and will recommend this article to be accepted.

---

> > > ### Author Response · Authors · 2024-11-23
> > >
> > > Thank you for your valuable suggestions to improve our manuscript and your positive feedback on the updated version. We remain optimistic about your evaluation and would be more than happy to discuss about any further questions or comments.

---

### Official Review · Reviewer_eopu · 2024-11-04

**Soundness:** 3
**Presentation:** 3
**Contribution:** 3
**Rating:** 6
**Confidence:** 2

**Summary:**

The paper developed Rainbow PO as a unified framework for direct preference optimization. Motivated by the challenge in comparing various xPO methods in literature, the authors began the study by analyzing more than 10 existing xPO methods and identified seven mathematically orthogonal components for mapping these methods. Among these components, they justified four of them as effective. Based on these findings, they proposed RainbowPO to combine these components. One notable novelty is that RainbowPO adopts a mixing reference policy mechanism. The authors conducted experiments to illustrate RainbowPO's superior performance over previous DPO variants.

**Strengths:**

1. The paper strived to provide a systematic and comprehensive study of existing preference optimization methods. The identification of the seven components leads to a structured framework for evaluating xPO methods, which is useful considering the large, and still growing number of xPO methods.

2. The paper contributed clarity and insights on how different components of an xPO method affect the method’s performance. The experiment evaluation offers interesting results illustrating the value of adding components to an xPO model.

**Weaknesses:**

There lack transparency and explanation on the identification of the seven components. It would be helpful to include further details on the identification process. Relevant questions include: Are these components necessary and sufficient to categorize current and potential new xPO models? Have similar components been applied or mentioned in literature on xPO? Are there any potential interactions among components that would affect the performance of xPO?

**Questions:**

Please answer the questions mentioned in ‘Weaknesses’.

---

> ### Author Response · Authors · 2024-11-16
>
> Thank you for providing useful comments to better improve the paper. We address your questions below:
>
> > There lack transparency and explanation on the identification of the seven components
>
> In our paper, we have comprehensively classified existing DPO methods into seven components, supported by an extensive literature review. We highlighted the significance of each component and thoughtfully integrated them into a unified objective, which we named RainbowPO. Additionally, we provide detailed descriptions of each component to emphasize their individual contributions.
>
> > Are these components necessary and sufficient to categorize current xPO models?
>
> We have done extensive literature review and experiments to identify components contributed by most existing DPO variants, and included only the helpful/performant components into our single cohesive objective in this paper. Thus we believe that our proposed objective **already characterize most of existing xPOs**.
>
>
> > Are these components necessary and sufficient to categorize potential new xPO models
>
> **For new/future xPOs**, our answer is uncertain, but we believe that our methods can clearly inspire the designs of new xPO algorithms and if one comes up with a new effective orthogonal component, they could easily improve RainbowPO by directly adding that component into a new objective. Our work is the first to exclusively **summarize** the effectiveness of components proposed to improve over DPO, including components already adopted (like length normalization, contextual scaling) and **new** components proposed by us (mixing reference policy), and fruitfully **combine** them.
>
> > Are there any potential interactions among components that would affect the performance of xPO?
>
> We would like to refer to some findings of the **interactions** between components in our experimental section (Sec. 4.1), in which we show that some components may not be complementary to each other in experiments, like length normalization and rejection sampling, though they are mathematically orthogonal.

---

> > ### Author Response · Authors · 2024-11-23
> >
> > Dear Reviewer eopu,
> >
> > We hope this message finds you well. We kindly want to check if you had a chance to review our rebuttal, and if you have any further questions or comments we can address to help with your evaluation. Thanks again for your efforts and suggestions in improving our manuscript.
> >
> > Sincerely, The authors

---

> > > ### Comment · Reviewer_eopu · 2024-11-24
> > >
> > > I would like to thank the authors for their detailed response to my comments and questions. The responses helped clarifying the paper's contributions to me, therefore, I have updated my scores accordingly.

---

### Official Review · Reviewer_Nycb · 2024-11-08

**Soundness:** 3
**Presentation:** 3
**Contribution:** 3
**Rating:** 6
**Confidence:** 4

**Summary:**

The paper extracts key components from various DPO variants and provides theoretical analysis for each component. It then establishes a unified formulation that encompasses most DPO variants. Empirically, the author conducts a series of ablation studies to analyze the effectiveness of each component in DPO variants and identifies the key components that contribute to metric gains. By combining these components, the author proposes a new variant of DPO called RainbowPO, which outperforms all other DPO variants.

**Strengths:**

1. This work is the first to provide a comprehensive analysis of the effectiveness of various DPO variants under the same setting, making it a valuable contribution to the community.

2. The study offers an in-depth examination of the components of DPO, supported by thorough experiments that validate the analysis, resulting in a solid and reliable evaluation.

3. By combining all effective components, this work successfully formulates a new variant of DPO and demonstrates its superior performance over existing algorithms, showcasing the potential for further advancements in the field.

**Weaknesses:**

1. The evaluation benchmark is limited to AlpacaEval, which may not provide sufficient evidence to support the claims regarding the effectiveness of each component and the advantages of the proposed RainbowPO. A more comprehensive evaluation across multiple benchmarks would be necessary to further validate these findings.

2. The advantage of introducing key components in RainbowPO lacks systematic explanation and remains at a purely empirical level. It is unclear whether the same advantages can be transferred to other tasks, highlighting the need for further investigation into the generalizability of these findings.

3. The proposed method introduces significant additional complexity compared to baseline methods, which may impact its practicality and scalability. Further research is needed to determine whether the benefits of RainbowPO outweigh the increased complexity it introduces.

**Questions:**

1. For eq (1), how to obtain the policy $\pi_\gamma$ in the practical setting?
2. What is the purpose of raising eq (12)? If left side is a lower bound then whether minimizing the lower bound can imply anything?

---

> ### Author Response · Authors · 2024-11-16
>
> Thank you for your positive review and constructive feedback. We appreciate your emphasis on the significance of our work and its potential impact on the field and future research. Below, we address your questions as follows:
>
> > the increased complexity it introduces
>
> Our RainbowPO method consolidates existing DPO methods from the literature into a single, streamlined formula. While it does introduce more hyperparameters compared to vanilla DPO (similar to other DPO variants) the insights gained from our empirical experiments provide a valuable starting point for tuning the model in future applications and extensions.
>
> > For eq (1), how to obtain the policy in the practical setting?
>
> To obtain a good policy $\pi_\gamma$, we first tune $\gamma$ as in SimPO, and then we tune the mixing hyperparameter $\alpha$. We have updated and included this part in the manuscript.
>
> > What is the purpose of raising eq (12)?
>
> Thank you for pointing this out. We have removed this part.

---

### Official Review · Reviewer_FkHK · 2024-11-08

**Soundness:** 3
**Presentation:** 2
**Contribution:** 3
**Rating:** 6
**Confidence:** 3

**Summary:**

This paper aims to provide a unified framework for direct preference optimization (DPO) methods which are becoming a popular and successful method to train or finetune generative language models, and in particular, for better alignment with human preferences. In DPO, we are given a dataset of triples consisting of a prompt and a pair of potential responses, one of which is known to be preferred over the other. The high level goal is to fine tune a large language model in a manner that maximizes the difference in the likelihood of the preferred and dispreferred responses. The performance of a model is measured by the win rate of responses of the model being preferred over the responses of another model over a collection of inputs as predicted by an automatic evaluator which is typically another generative language model trained to predict which of two responses a human annotator would prefer.

The paper surveys, then carefully analyzes the objective functions of different algorithms in a growing family of DPO like models, identifies their components and then formulates an objective function for their proposed RainbowPO method which incorporates the components and techniques of previous DPO methods. The contribution of the different components of the objective function can be tuned smoothly using hyperparameters. The paper then goes to evaluate the effectiveness and contributions of each component to the performance of DPO methods.

After rebuttal:
I am increasing my score from 5 to 6. The authors have addressed my questions. Overall, I lean towards acceptance if there is room to accommodate it.

**Strengths:**

- A careful analysis of different DPO methods, identifying their components and understanding the contribution of components of different DPO methods to performance is a worthwhile endeavor. The main technical contribution is a careful analysis and rewriting of the objective functions of recently proposed DPO methods and unifying them as a single objective function.
- Another contribution is the proposed the mixed reference policy obtained by mixing a fine tuned policy (model) with a policy that maintains a certain margin or odds ratio of the likelihood of the preferred to dispreferred responses.
- I believe the main conceptual contribution is the development of common frameworks and pipelines that allow researchers and practitioners to evaluate different components and techniques for DPO. As such, the paper is well motivated and the findings of this paper and follow on works that attempt to perform such evaluations will be of significance as alignment of generative language models is an important problem and existing reinforcement learning based approaches are computationally expensive in comparison to DPO.
- The main finding that length normalization, policy mixing and contextual scaling contributed towards increasing performance are a potentially interesting takeaway that may be significant to practitioners.

**Weaknesses:**

- The paper is overall poorly written with several missing explanations and gaps in the logic. While it is possible to guess what is meant, a reader unfamiliar with the area will likely find it very hard to follow.
- At a high level, the organization of the paper makes it hard to read as the text on the background and proposed RainbowPO model frequently references results in the later experimental section which distracts from understanding the components of the DPO objective functions.
- The technical contributions and novelty are limited beyond the creation of the unified objective function and evaluation pipeline.

Some examples of issues with the writing:
- line 153: ... there lacks a work .. please consider revising. This is just one minor example. There are several more throughout the paper and they are too numerous to list. Please consider performing a thorough reading of the paper.
- please consider elaborating on the verbosity issue and how it may be potentially a way to game the evaluator
- different places use y_l and y^l. Is this a typo? Similar issues exist elsewhere. Preferably please keep the usage of subscript and superscript consistent. For example, see in (10) the use of \pi_\alpha and \pi_{ref}^\alpha. Consider e.g. using \pi_\alpha, \pi_\alpha^{ref} and \pi*{ref} as appropriate instead
- abbreviation LC (length controlled) is not defined or explained anywhere making it hard to understand results in Section 4.1
- in general, it would be nice to have a more clear discussion of the evaluation framework and evaluation criteria. As it stands, the paper assumes too much background knowledge e.g. about AlpacaEval.

**Questions:**

- Could you please elaborate on the evaluation? What was the base model against which the win rate was  evaluated in Section 4.
- Could you please elaborate on the use of link functions?
- Could you please elaborate on the mixing reference policy results in Section 4? What are the policies being mixed? How is the "good" policy \pi_\gamma obtained?

---

> ### Author Response · Authors · 2024-11-16
> **Improved clarity and more details added to the manuscript**
>
> Thank you for providing detailed feedback and suggestions on our manuscript. Also thank you for emphasizing the significance of our work and its potential impact on the field and future research. Here we answer your questions and suggestions as following:
>
> > Revising manuscript for typos and more explanations
>
> We appreciate your detailed suggestions for improving the quality of our writing. We have thoroughly revised our manuscript to ensure that all sections are clearly explained or appropriately referenced.
>
> > please consider elaborating on the verbosity issue and how it may be potentially a way to game the evaluator
>
> In Section 3.1 of the updated manuscript, now highlighted in blue, we have included a detailed description of the verbosity of DPO-aligned models and provided evidence of how LLM-as-a-judge may exhibit length bias.
>
> > different places use $y_l$ and $y^l$
>
> Thank you for bringing this to our attention; it was indeed a typo. We have corrected it and ensured that the terminology is consistent throughout the document.
>
> > the use of $\pi_\alpha$ and $\pi_{ref}^\alpha$. Consider e.g. using $\pi_\alpha$, $\pi_\alpha^{ref}$ and $\pi^{ref}$ as appropriate instead
>
> The notation $\pi_\alpha$ (with $\alpha$ as a subscript) represents the policy, whereas $\pi_{ref}^\alpha$ (with $\alpha$ as a superscript) indicates the likelihood raised to the power of $\alpha$. We have updated this notation in the relevant section and highlighted the changes in blue for your convenience during review.
>
> > Could you please elaborate on the evaluation?
>
> We have expanded the discussion of AlpacaEval in Section 4, with the changes highlighted in blue for easy reference. The reference/base answer is generated by GPT-4 turbo. Additionally, we have included descriptions, the logic behind the metrics (Win Rates and Length controlled Win Rates), and the generation hyperparameters of the models in Appendix E.1.
>
> > Could you please elaborate on the use of link functions?
>
> In the “Link Function” section of Section 3, we have included the link functions used by IPO and SLiC. Additionally, we have provided a more detailed categorization of XPOs in Appendix C, where the link functions can be directly observed from the XPOs’ loss objectives. Based on our experimental results, we discovered that the logistic function in DPO yields the best performance, so we have adopted this link function in our RainbowPO.
>
> > Could you please elaborate on the mixing reference policy results in Section 4? What are the policies being mixed? How is the "good" policy $\pi_\gamma$ obtained?
>
> We have updated the Reference Policy Section with more details. The mixed policies are the SFT policy and an “ideal” policy related to the margin denoted by gamma. To obtain a good policy $\pi_\gamma$, we first tune $\gamma$ as in SimPO, and then we tune the mixing hyperparameter $\alpha$. We have also included this part in the manuscript.

---

> > ### Author Response · Authors · 2024-11-23
> >
> > Dear Reviewer FkHK,
> >
> > We hope this message finds you well. We kindly want to check if you had a chance to review our rebuttal, and if you have any further questions or comments we can address to help with your evaluation. Thanks again for your efforts and suggestions in improving our manuscript.
> >
> > Sincerely, The authors

---

> ### Author Response · Authors · 2024-11-25
> **Follow-Up on Rebuttal and Updated Manuscript**
>
> Dear Reviewer FkHK,
>
> We hope this message finds you well. As the discussion period is nearing its conclusion, we wanted to kindly follow up to check if you’ve had a chance to review our rebuttal and the updated version of our paper.
>
> If we have successfully addressed all your concerns, we would sincerely appreciate it if you could consider updating your scores accordingly. We are grateful for your efforts and constructive feedback, which helped us improve our manuscript.
>
> If you have any additional questions or comments, we would be more than happy to discuss and address them within the remaining discussion period.
>
> Thank you once again for your time and support.
>
> Best regards, The authors

---

> > ### Comment · Reviewer_FkHK · 2024-11-25
> >
> > Thank you for your response. It addresses my questions and I do not have any further questions. I have updated my scores.

---

### Meta-Review · Area_Chair_q5mh · 2024-12-30

**Metareview:**

This work reviews several improvements over DPO and identifies "orthogonal" complementary components whose contribution to task alignment has not been thoroighly examined.  Their proposed training procedure, RainbowDPO, combines aspects of multiple approaches, which can be parameterized by multiple hyperparameters.  The authors investigate the effects of each of these mechanisms, finding that length normalization, reference policy mixing and contextual scaling then to produce the largest improvements in terms of automated eval (AlpacaEval2) metrics and length.

Reviewers found the paper well-written and commended the authors' effort at summarizing and evaluating recent DPO methods within the same framework. Reviewers found the insights and unified parameterization of DPO to be valuable.  After the rebuttal stage, all reviewers gave the paper a "marginally above acceptance threshold" rating and advocated for acceptance.

There were a few weaknesses pointed out by the reviewers which I hope can be addressed in a future revision.  In particular, Nycb calls out that the evaluation benchmark is limited to AlpacaEval2 scores for a fixed set of tasks, so it is not clear how the findings generalize across different tasks.  Evaluation across multiple tasks can help with understanding generalization here.

The authors' method for identifying important hyperparameters (and manual hyperparameter optimization) is ad hoc and could easily be evaluated more rigorously via sensitivity analysis (e.g., via sobol indicies or derivative-based methods), and optimal HPs can be found efficiently via Bayesian optimization (or multi-objective BO if targeting eval metrics and length).  Such optimization is easy to do with off-the-shelf tools.  It also appears that the authors only performed HPO for RainbowPO and not other methods; performing HPO for each individual method (of course, using a valid train/test/validation split) is crucial for making comparisons between methods.  The authors appear to only consider a limited number of training epochs (1 and 3), which to the AC feels fairly arbitrary.  Since there is heterogeneity between how the different methods/HPs, it might be beneficial to consider this as a tunable HP.

**Additional Comments On Reviewer Discussion:**

Reviewers asked for clarifying questions, and in some cases requested additional baselines.  The authors addressed many issues raised via their rebuttal + revision, after which point reviewers raised their scores to all 6s.  This work satisfied both junior and expert reviewers and so I vouch for accept.

---

### Decision · Program_Chairs · 2025-01-22

Accept (Poster)